# CEIL: Generalized Contextual Imitation Learning

**Jinxin Liu**[1,2*]    **Li He**[1*]    **Yachen Kang**[1,2]    **Zifeng Zhuang**[1,2]

**Donglin Wang**[1,4]    **Huazhe Xu**[3,5,6]

[1]Westlake University   [2]Zhejiang University   [3]Tsinghua University
[4]Westlake Institute for Advanced Study   [5]Shanghai Qi Zhi Institute   [6]Shanghai AI Lab

## Abstract

In this paper, we present **C**ont**E**xtual **I**mitation **L**earning (CEIL), a general and broadly applicable algorithm for imitation learning (IL). Inspired by the formulation of hindsight information matching, we derive CEIL by explicitly learning a hindsight embedding function together with a contextual policy using the hindsight embeddings. To achieve the expert matching objective for IL, we advocate for optimizing a contextual variable such that it biases the contextual policy towards mimicking expert behaviors. Beyond the typical learning from demonstrations (LfD) setting, CEIL is a generalist that can be effectively applied to multiple settings including: 1) learning from observations (LfO), 2) offline IL, 3) cross-domain IL (mismatched experts), and 4) one-shot IL settings. Empirically, we evaluate CEIL on the popular MuJoCo tasks (online) and the D4RL dataset (offline). Compared to prior state-of-the-art baselines, we show that CEIL is more sample-efficient in most online IL tasks and achieves better or competitive performances in offline tasks.

## 1   Introduction

Imitation learning (IL) allows agents to learn from expert demonstrations. Initially developed with a supervised learning paradigm [58, 63], IL can be extended and reformulated with a general expert matching objective, which aims to generate policies that produce trajectories with low distributional distances to expert demonstrations [30]. This formulation allows IL to be extended to various new settings: 1) online IL where interactions with the environment are allowed, 2) learning from observations (LfO) where expert actions are absent, 3) offline IL where agents learn from limited expert data and a fixed dataset of sub-optimal and reward-free experience, 4) cross-domain IL where the expert demonstrations come from another domain (*i.e.*, environment) that has different transition dynamics, and 5) one-shot IL which expects to recover the expert behaviors when only one expert trajectory is observed for a new IL task.

Modern IL algorithms introduce various designs or mathematical principles to cater to the expert matching objective in a specific scenario. For example, the LfO setting requires particular considerations regarding the absent expert actions, *e.g.*, learning an inverse dynamics function [5, 65]. Besides, out-of-distribution issues in offline IL require specialized modifications to the learning objective, such as introducing additional policy/value regularization [32, 72]. However, such a methodology, designing an individual formulation for each IL setting, makes it difficult to scale up a specific IL algorithm to more complex tasks beyond its original IL setting, *e.g.*, online IL methods often suffer severe performance degradation in offline IL settings. Furthermore, realistic IL tasks are often not subject to a particular IL setting but consist of a mixture of them. For example, we may have access

---

*Equal contributions. Corresponding author: Donglin Wang <wangdonglin@westlake.edu.cn>

Table 1: A coarse summary of IL methods demonstrating 1) different expert data modalities they can handle (learning from *demonstrations* or *observations*), 2) disparate task settings they consider (learning from *online* environment interactions or pre-collected *offline* static dataset), 3) the specific *cross-domain* setting they assume (the transition dynamics between the learning environment and that of the expert behaviors are different), and 4) the unique *one-shot* merit they desire (the learned policy is capable of one-shot transfer to new imitation tasks). We highlight that our contextual imitation learning (CEIL) method can naturally be applied to all the above IL settings.

| | Expert data | | Task setting | | Cross-domain | One-shot |
|---|---|---|---|---|---|---|
| | LfD | LfO | Online | Offline | | |
| S-on-LfD [9, 13, 21, 30, 38, 52, 57, 61, 77] | ✔ | ✗ | ✔ | ✗ | ✗ | ✗ |
| S-on-LfO [7, 54, 65, 66, 75] | ✔ | ✔ | ✔ | ✗ | ✗ | ✗ |
| S-off-LfD [19, 32, 33, 39, 55, 70, 72, 73] | ✔ | ✗ | ✗ | ✔ | ✗ | ✗ |
| S-off-LfO [78] | ✔ | ✔ | ✗ | ✔ | ✗ | ✗ |
| C-on-LfD [18, 69, 79] | ✔ | ✗ | ✔ | ✗ | ✔ | ✗ |
| C-on-LfO [20, 25, 26, 48, 59, 60] | ✔ | ✔ | ✔ | ✗ | ✔ | ✗ |
| C-off-LfD [34] | ✔ | ✗ | ✗ | ✔ | ✔ | ✗ |
| C-off-LfO [56, 68] | ✔ | ✔ | ✗ | ✔ | ✔ | ✗ |
| S-on/off-LfO [28] | ✔ | ✔ | ✔ | ✔ | ✗ | ✗ |
| Online one-shot [14, 16, 40] | ✔ | ✗ | ✔ | ✗ | ✗ | ✔ |
| Offline one-shot [24, 71] | ✔ | ✗ | ✗ | ✔ | ✗ | ✔ |
| **CEIL (ours)** | ✔ | ✔ | ✔ | ✔ | ✔ | ✔ |

to both demonstrations and observation-only data in offline robot tasks; however, it could require significant effort to adapt several specialized methods to leverage such mixed/hybrid data. Hence, a problem naturally arises: *How can we accommodate various design requirements of different IL settings with a general and practically ready-to-deploy IL formulation?*

Hindsight information matching, a task-relabeling paradigm in reinforcement learning (RL), views control tasks as analogous to a general sequence modeling problem, with the goal to produce a sequence of actions that induces high returns [12]. Its generality and simplicity enable it to be extended to both online and offline settings [17, 42]. In its original RL context, an agent directly uses known extrinsic rewards to bias the hindsight information towards task-related behaviors. However, when we attempt to retain its generality in IL tasks, how to bias the hindsight towards expert behaviors remains a significant barrier as the extrinsic rewards are missing.

To design a general IL formulation and tackle the above problems, we propose **C**ont**E**xtual **I**mitation **L**earning (CEIL), which readily incorporates the hindsight information matching principle within a bi-level expert matching objective. In the inner-level optimization, we explicitly learn a hindsight embedding function to deal with the challenges of unknown rewards. In the outer-level optimization, we perform IL expert matching via inferring an optimal embedding (*i.e.*, hindsight embedding biasing), replacing the naive reward biasing in hindsight. Intuitively, we find that such a bi-level objective results in a spectrum of expert matching objectives from the embedding space to the trajectory space. To shed light on the applicability and generality of CEIL, we instantiate CEIL to various IL settings, including online/offline IL, LfD/LfO, cross-domain IL, and one-shot IL settings.

In summary, this paper makes the following contributions: 1) We propose a bi-level expert matching objective ContExtual Imitation Learning (CEIL), inheriting the spirit of hindsight information matching, which decouples the learning policy into a contextual policy and an optimal embedding. 2) CEIL exhibits high generality and adaptability and can be instantiated over a range of IL tasks. 3) Empirically, we conduct extensive empirical analyses showing that CEIL is more sample-efficient in online IL and achieves better or competitive results in offline IL tasks.

## 2 Related Work

Recent advances in decision-making have led to rapid progress in IL settings (Table 1), from typical learning from demonstrations (LfD) to learning from observations (LfO) [7, 9, 35, 54, 62, 66], from online IL to offline IL [11, 15, 33, 53, 73], and from single-domain IL to cross-domain IL [34, 48, 56, 68]. Targeting a specific IL setting, individual works have shown their impressive

ability to solve the exact IL setting. However, it is hard to retrain their performance in new unprepared IL settings. In light of this, it is tempting to consider how we can design a general and broadly applicable IL method. Indeed, a number of prior works have studied part of the above IL settings, such as offline LfO [78], cross-domain LfO [48, 60], and cross-domain offline IL [56]. While such works demonstrate the feasibility of tackling multiple IL settings, they still rely on standard online/offline RL algorithmic advances to improve performance [25, 32, 44, 47, 50, 51, 55, 72, 76]. Our objective diverges from these works, as we strive to minimize the reliance on the RL pipeline by replacing it with a simple supervision objective, thus avoiding the dependence on the choice of RL algorithms.

Our approach to IL is most closely related to prior hindsight information-matching methods [2, 8, 24, 49], both learning a contextual policy and using a contextual variable to guide policy improvement. However, these prior methods typically require additional mechanisms to work well, such as extrinsic rewards in online RL [4, 42, 64] or a handcrafted target return in offline RL [12, 17]. Our method does not require explicit handling of these components. By explicitly learning an embedding space for both expert and suboptimal behaviors, we can bias the contextual policy with an inferred optimal embedding (contextual variable), thus avoiding the need for explicit reward biasing in prior works. Our method also differs from most prior offline transformer-based RL/IL algorithms that explicitly model a long sequence of transitions [10, 12, 31, 36, 43, 71]. We find that simple fully-connected networks can also elicit useful embeddings and guide expert behaviors when conditioned on a well-calibrated embedding. In the context of the recently proposed prompt-tuning paradigm in large language tasks or multi-modal tasks [27, 45, 74], our method can be interpreted as a combination of IL and prompting-tuning, with the main motivation that we tune the prompt (the optimal contextual variable) with an expert matching objective in IL settings.

## 3 Background

Before discussing our method, we briefly introduce the background for IL, including learning from demonstrations (LfD), learning from observations (LfO), online IL, offline IL, and cross-domain settings in Section 3.1, and introduce the hindsight information matching in Section 3.2.

### 3.1 Imitation Learning

Consider a control task formulated as a discrete-time Markov decision process (MDP)[2] $\mathcal{M} = \{\mathcal{S}, \mathcal{A}, \mathcal{T}, r, \gamma, p_0\}$, where $\mathcal{S}$ is the state (observation) space, $\mathcal{A}$ is the action space, $\mathcal{T} : \mathcal{S} \times \mathcal{A} \times \mathcal{S} \to \mathbb{R}$ is the transition dynamics function, $r : \mathcal{S} \times \mathcal{A} \to \mathbb{R}$ is the reward function, $\gamma$ is the discount factor, and $p_0$ is the distribution of initial states. The goal in a reinforcement learning (RL) control task is to learn a policy $\pi_\theta(\mathbf{a}|\mathbf{s})$ maximizing the expected sum of discounted rewards $\mathbb{E}_{\pi_\theta(\tau)} \left[ \sum_{t=0}^{T-1} \gamma^t r(\mathbf{s}_t, \mathbf{a}_t) \right]$, where $\tau := \{\mathbf{s}_0, \mathbf{a}_0, \cdots, \mathbf{s}_{T-1}, \mathbf{a}_{T-1}\}$ denotes the trajectory and the generated trajectory distribution $\pi_\theta(\tau) = p_0(\mathbf{s}_0)\pi_\theta(\mathbf{a}_0|\mathbf{s}_0) \prod_{t=1}^{T-1} \pi_\theta(\mathbf{a}_t|\mathbf{s}_t)\mathcal{T}(\mathbf{s}_t|\mathbf{s}_{t-1}, \mathbf{a}_{t-1})$.

In IL, the ground truth reward function (*i.e.*, $r$ in $\mathcal{M}$) is not observed. Instead, we have access to a set of demonstrations (or observations) $\{\tau | \tau \sim \pi_E(\tau)\}$ that are collected by an unknown expert policy $\pi_E(\mathbf{a}|\mathbf{s})$. The goal of IL tasks is to recover a policy that matches the corresponding expert policy. From the mathematical perspective, IL achieves the plain expert matching objective by minimizing the divergence of trajectory distributions between the learner and the expert:

$$\min_{\pi_\theta} \ D(\pi_\theta(\tau), \pi_E(\tau)), \tag{1}$$

where $D$ is a distance measure. Meanwhile, we emphasize that the given expert data $\{\tau | \tau \sim \pi_E(\tau)\}$ may not contain the corresponding expert actions. Thus, in this work, we consider two IL cases where the given expert data $\tau$ consists of a set of state-action demonstrations $\{(\mathbf{s}_t, \mathbf{a}_t, \mathbf{s}_{t+1})\}$ (learning from demonstrations, LfD), as well as a set of state-only transitions $\{(\mathbf{s}_t, \mathbf{s}_{t+1})\}$ (learning from observations, LfO). *When it is clear from context, we abuse notation $\pi_E(\tau)$ to denote both demonstrations in LfD and observations in LfO for simplicity.*

Besides, we can also divide IL settings into two orthogonal categories: online IL and offline IL. In online IL, the learning policy $\pi_\theta$ can interact with the environment and generate online trajectories $\tau \sim \pi_\theta(\tau)$. In offline IL, the agent cannot interact with the environment but has access to an offline

---

[2]In this paper, we use environment and MDP interchangeably, and use state and observation interchangeably.

static dataset $\{\boldsymbol{\tau} | \boldsymbol{\tau} \sim \pi_\beta(\boldsymbol{\tau})\}$, collected by some unknown (sub-optimal) behavior policies $\pi_\beta$. By leveraging the offline data $\{\pi_\beta(\boldsymbol{\tau})\} \cup \{\pi_E(\boldsymbol{\tau})\}$ without any interactions with the environment, the goal of offline IL is to learn a policy recovering the expert behaviors (demonstrations or observations) generated by $\pi_E$. Note that, in contrast to the typical offline RL problem [46], the offline data $\{\pi_\beta(\boldsymbol{\tau})\}$ in offline IL does not contains any reward signal.

**Cross-domain IL.** Beyond the above two IL branches (online/offline and LfD/LfO), we can also divide IL into: 1) single-domain IL and 2) cross-domain IL, where 1) the single-domain IL assumes that the expert behaviors are collected in the same MDP in which the learning policy is to be learned, and 2) the cross-domain IL studies how to imitate expert behaviors when discrepancies exist between the expert and the learning MDPs (*e.g.*, differing in their transition dynamics or morphologies).

## 3.2 Hindsight Information Matching

In typical goal-conditioned RL problems, hindsight experience replay (HER) [3] proposes to leverage the rich repository of the failed experiences by replacing the desired (true) goals of training trajectories with the achieved goals of the failed experiences:

$$\texttt{Alg}(\pi_\theta; \boldsymbol{g}, \boldsymbol{\tau_g}) \rightarrow \texttt{Alg}(\pi_\theta; f_{\text{HER}}(\boldsymbol{\tau_g}), \boldsymbol{\tau_g}),$$

where the learner $\texttt{Alg}(\pi_\theta; \cdot, \cdot)$ could be any RL methods, $\boldsymbol{\tau_g} \sim \pi_\theta(\boldsymbol{\tau_g} | \boldsymbol{g})$ denotes the trajectory generated by a goal-conditioned policy $\pi_\theta(\mathbf{a}_t | \mathbf{s}_t, \boldsymbol{g})$, and $f_{\text{HER}}$ denotes a pre-defined (hindsight information extraction) function, *e.g.*, returning the last state in trajectory $\boldsymbol{\tau_g}$.

HER can also be applied to the (single-goal) reward-driven online/offline RL tasks, setting the return (sum of the discounted rewards) of a trajectory as an implicit goal for the corresponding trajectory. Thus, *we can reformulate the (single-goal) reward-driven RL task*, learning policy $\pi_\theta(\mathbf{a}_t | \mathbf{s}_t)$ that maximize the return, *as a multi-goal RL task*, learning a return-conditioned policy $\pi_\theta(\mathbf{a}_t | \mathbf{s}_t, \cdot)$ that maximize the following log-likelihood:

$$\max_{\pi_\theta} \mathbb{E}_{\mathcal{D}(\boldsymbol{\tau})} \left[ \log \pi_\theta(\mathbf{a} | \mathbf{s}, f_{\text{R}}(\boldsymbol{\tau})) \right], \tag{2}$$

where $f_{\text{R}}(\boldsymbol{\tau})$ denotes the return of trajectory $\boldsymbol{\tau}$. At test, we can then condition the contextual policy $\pi_\theta(\mathbf{a} | \mathbf{s}, \cdot)$ on a desired target return. In offline RL, the empirical distribution $\mathcal{D}(\boldsymbol{\tau})$ in Equation 2 can be naturally set as the offline data distribution; in online RL, $\mathcal{D}(\boldsymbol{\tau})$ can be set as the replay/experience buffer, and will be updated and biased towards trajectories that have high expected returns.

Intuitively, biasing the sampling distribution ($\mathcal{D}(\boldsymbol{\tau})$ towards higher returns) leads to *an implicit policy improvement operation*. However, such an operator is non-trivial to obtain in the IL problem, where we do not have access to a pre-defined function $f_{\text{R}}(\boldsymbol{\tau})$ to bias the learning policy towards recovering the given expert data $\{\pi_E(\boldsymbol{\tau})\}$ (demonstrations or observations).

## 4 Method

In this section, we will formulate IL as a bi-level optimization problem, which will allow us to derive our method, contextual imitation learning (CEIL). Instead of attempting to train the learning policy $\pi_\theta(\mathbf{a} | \mathbf{s})$ with the plain expert matching objective (Equation 1), our approach introduces an additional contextual variable $\mathbf{z}$ for a contextual IL policy $\pi_\theta(\mathbf{a} | \mathbf{s}, \cdot)$. The main idea of CEIL is to learn a contextual policy $\pi_\theta(\mathbf{a} | \mathbf{s}, \mathbf{z})$ and an optimal contextual variable $\mathbf{z}^*$ such that the given expert data (demonstrations in LfD or observations in LfO) can be recovered by the learned $\mathbf{z}^*$-conditioned policy $\pi_\theta(\mathbf{a} | \mathbf{s}, \mathbf{z}^*)$. We begin by describing the overall framework of CEIL in Section 4.1, and make a connection between CEIL and the plain expert matching objective in Section 4.2, which leads to a practical implementation under various IL settings in Section 4.3.

### 4.1 Contextual Imitation Learning (CEIL)

Motivated by the hindsight information matching in online/offline RL (Section 3.2), we propose to learn a general hindsight embedding function $f_\phi$, which encodes trajectory $\boldsymbol{\tau}$ (with window size $T$) into a latent variable $\mathbf{z} \in \mathcal{Z}$, $|\mathcal{Z}| \ll T * |\mathcal{S}|$. Formally, we learn the embedding function $f_\phi$ and a corresponding contextual policy $\pi_\theta(\mathbf{a} | \mathbf{s}, \mathbf{z})$ by minimizing the trajectory self-consistency loss:

$$\pi_\theta, f_\phi = \min_{\pi_\theta, f_\phi} -\mathbb{E}_{\mathcal{D}(\boldsymbol{\tau})} \left[ \log \pi_\theta(\boldsymbol{\tau} | f_\phi(\boldsymbol{\tau})) \right] = \min_{\pi_\theta, f_\phi} -\mathbb{E}_{\boldsymbol{\tau} \sim \mathcal{D}(\boldsymbol{\tau})} \mathbb{E}_{(\mathbf{s}, \mathbf{a}) \sim \boldsymbol{\tau}} \left[ \log \pi_\theta(\mathbf{a} | \mathbf{s}, f_\phi(\boldsymbol{\tau})) \right], \tag{3}$$

where in the online setting, we sample trajectory $\boldsymbol{\tau}$ from buffer $\mathcal{D}(\boldsymbol{\tau})$, known as the experience replay buffer in online RL; in the offline setting, we sample trajectory $\boldsymbol{\tau}$ directly from the given offline data.

If we can ensure that the learned contextual policy $\pi_\theta$ and the embedding function $f_\phi$ are accurate on the empirical data $\mathcal{D}(\boldsymbol{\tau})$, then we can convert the IL policy optimization objective (in Equation 1) into a bi-level expert matching objective:

$$\min_{\mathbf{z}^*} \ D(\pi_\theta(\boldsymbol{\tau}|\mathbf{z}^*), \pi_E(\boldsymbol{\tau})), \tag{4}$$

$$\text{s.t. } \pi_\theta, f_\phi = \min_{\pi_\theta, f_\phi} -\mathbb{E}_{\mathcal{D}(\boldsymbol{\tau})}\left[\log \pi_\theta(\boldsymbol{\tau}|f_\phi(\boldsymbol{\tau}))\right] - \mathcal{R}(f_\phi), \text{ and } \mathbf{z}^* \in f_\phi \circ \mathtt{supp}(\mathcal{D}), \tag{5}$$

where $\mathcal{R}(f_\phi)$ is an added regularization over the embedding function (we will elaborate on it later), and $\mathtt{supp}(\mathcal{D})$ denotes the support of the trajectory distribution $\{\boldsymbol{\tau}|\mathcal{D}(\boldsymbol{\tau}) > 0\}$. Here $f_\phi$ is employed to map the trajectory space to the latent variable space ($\mathcal{Z}$). Intuitively, by optimizing Equation 4, we expect the induced trajectory distribution of the learned $\pi_\theta(\mathbf{a}|\mathbf{s}, \mathbf{z}^*)$ will match that of the expert. However, in the offline IL setting, the contextual policy can not interact with the environment. If we directly optimize the expert matching objective (Equation 4), such an objective can easily exploit generalization errors in the contextual policy model to infer a mistakenly overestimated $\mathbf{z}^*$ that achieves low expert-matching loss but does not preserve the trajectory self-consistency (Equation 3). Therefore, we formalize CEIL into a bi-level optimization problem, where, in Equation 5, we explicitly constrain the inferred $\mathbf{z}^*$ lies in the ($f_\phi$-mapped) support of the training trajectory distribution.

At a high level, CEIL decouples the learning policy into two parts: an expressive contextual policy $\pi_\theta(\mathbf{a}|\mathbf{s}, \cdot)$ and an optimal contextual variable $\mathbf{z}^*$. By comparing CEIL with the plain expert matching objective, $\min_{\pi_\theta} D(\pi_\theta(\boldsymbol{\tau}), \pi_E(\boldsymbol{\tau}))$, in Equation 1, we highlight two merits: 1) CEIL's expert matching loss (Equation 4) does not account for updating $\pi_\theta$ and is only incentivized to update the low-dimensional latent variable $\mathbf{z}^*$, which enjoys efficient parameter learning similar to the prompt tuning in large language models [74], and 2) we learn $\pi_\theta$ by simply performing supervised regression (Equation 5), which is more stable compared to vanilla inverse-RL/adversarial-IL methods.

## 4.2 Connection to the Plain Expert Matching Objective

To gain more insight into Equation 4 that captures the quality of IL (the degree of similarity to the expert data), we define $D(\cdot, \cdot)$ as the sum of reverse KL and forward KL divergence[3], *i.e.*, $D(q, p) = D_{\mathrm{KL}}(q\|p) + D_{\mathrm{KL}}(p\|q)$, and derive an alternative form for Equation 4:

$$\arg\min_{\mathbf{z}^*} \ D(\pi_\theta(\boldsymbol{\tau}|\mathbf{z}^*), \pi_E(\boldsymbol{\tau})) = \arg\max_{\mathbf{z}^*} \ \underbrace{\mathcal{I}(\mathbf{z}^*; \boldsymbol{\tau}_E) - \mathcal{I}(\mathbf{z}^*; \boldsymbol{\tau}_\theta)}_{\mathcal{J}_{\mathrm{MI}}} - \underbrace{D(\pi_\theta(\boldsymbol{\tau}), \pi_E(\boldsymbol{\tau}))}_{\mathcal{J}_D}, \tag{6}$$

where $\mathcal{I}(\mathbf{x}; \mathbf{y})$ denotes the mutual information (MI) between $\mathbf{x}$ and $\mathbf{y}$, which measures the predictive power of $\mathbf{y}$ on $\mathbf{x}$ (or vice-versa), the latent variables are defined as $\boldsymbol{\tau}_E := \boldsymbol{\tau} \sim \pi_E(\boldsymbol{\tau})$, $\boldsymbol{\tau}_\theta := \boldsymbol{\tau} \sim p(\mathbf{z}^*)\pi_\theta(\boldsymbol{\tau}|\mathbf{z}^*)$, and $\pi_\theta(\boldsymbol{\tau}) = \mathbb{E}_{\mathbf{z}^*}\left[\pi_\theta(\boldsymbol{\tau}|\mathbf{z}^*)\right]$.

Intuitively, the second term $\mathcal{J}_D$ on RHS of Equation 6 is similar to the plain expert matching objective in Equation 1, except that here we optimize a latent variable $\mathbf{z}^*$ over this objective. Regarding the MI terms $\mathcal{J}_{\mathrm{MI}}$, we can interpret the maximization over $\mathcal{J}_{\mathrm{MI}}$ as an implicit policy improvement, which incentivizes the optimal latent variable $\mathbf{z}^*$ for having high predictive power of the expert data $\boldsymbol{\tau}_E$ and having low predictive power of the non-expert data $\boldsymbol{\tau}_\theta$.

Further, we can rewrite the MI term ($\mathcal{J}_{\mathrm{MI}}$ in Equation 6) in terms of the learned embedding function $f_\phi$, yielding an approximate embedding inference objective $\mathcal{J}_{\mathrm{MI}(f_\phi)}$:

$$\mathcal{J}_{\mathrm{MI}} = \mathbb{E}_{\pi_E(\mathbf{z}^*, \boldsymbol{\tau}_E)} \log p(\mathbf{z}^*|\boldsymbol{\tau}_E) - \mathbb{E}_{\pi_\theta(\mathbf{z}^*, \boldsymbol{\tau}_\theta)} \log p(\mathbf{z}^*|\boldsymbol{\tau}_\theta)$$
$$\approx \mathbb{E}_{p(\mathbf{z}^*)\pi_E(\boldsymbol{\tau}_E)\pi_\theta(\boldsymbol{\tau}_\theta|\mathbf{z}^*)} \left[-\|\mathbf{z}^* - f_\phi(\boldsymbol{\tau}_E)\|^2 + \|\mathbf{z}^* - f_\phi(\boldsymbol{\tau}_\theta)\|^2\right] \triangleq \mathcal{J}_{\mathrm{MI}(f_\phi)},$$

where we approximate the logarithmic predictive power of $\mathbf{z}^*$ on $\boldsymbol{\tau}$ with $-\|\mathbf{z}^* - f_\phi(\boldsymbol{\tau})\|^2$, by taking advantage of the learned embedding function $f_\phi$ in Equation 5.

---

[3] $D_{\mathrm{KL}}(p\|q) := \mathbb{E}_{p(\mathbf{x})}\left[\log \frac{p(\mathbf{x})}{q(\mathbf{x})}\right]$ denotes the (forward) KL divergences. It is well known that reverse KL ensures that the learned distribution is mode-seeking and forward KL exhibits a mode-covering behavior [37]. For analysis purposes, here we define $D(\cdot, \cdot)$ as the sum of reverse KL and forward KL, and set the weights of both reverse KL and forward KL to 1.

**Algorithm 1** Training CEIL: Online *or* Offline IL Setting

**Require:** Expert demonstrations $\{\pi_E(\boldsymbol{\tau})\}$, empty buffer $\mathcal{D}$ for online IL *or* reward-free offline data $\mathcal{D}$ for offline IL, training iteration $K$, and batch size $N$.
1: Initialize contextual policy $\pi_\theta(\mathbf{a}|\mathbf{s}, \cdot)$, embedding function $f_\phi(\mathbf{z}|\boldsymbol{\tau})$, and latent variable $\mathbf{z}^*$.
2: **for** $k = 1, \cdots, K$ **do**
3:    (Online only) Run policy $\pi_\theta(\mathbf{a}|\mathbf{s}, \mathbf{z}^*)$ in environment and store experience into buffer $\mathcal{D}$.
4:    Sample a batch of data $\{\boldsymbol{\tau}\}_1^n$ from $\mathcal{D}$ for online IL *or* $\mathcal{D}$ for offline IL.
5:    Learn $\pi_\theta$ and $f_\phi$ over sampled $\{\boldsymbol{\tau}\}_1^n$ using the trajectory self-consistency loss.
6:    Update $\mathbf{z}^*$ and $f_\phi$ over sampled $\{\boldsymbol{\tau}\}_1^n$ by maximizing $\mathcal{J}_{\mathrm{MI}(f_\phi)} - \alpha \mathcal{J}_D$.
7:    (Offline only) Update $\mathbf{z}^*$ by minimizing $\mathcal{R}(\mathbf{z}^*)$.   # eliminating the offline OOD issues.
8: **end for**

**Return:** the learned contextual policy $\pi_\theta(\mathbf{a}|\mathbf{s}, \cdot)$ and the optimal latent variable $\mathbf{z}^*$.

By maximizing $\mathcal{J}_{\mathrm{MI}(f_\phi)}$, the learned optimal $\mathbf{z}^*$ will be induced to converge towards the embeddings of expert data and avoid trivial solutions (as shown in Figure 1). Intuitively, $\mathcal{J}_{\mathrm{MI}(f_\phi)}$ can also be thought of as an instantiation of contrastive loss, which manifests two facets we consider significant in IL:

1) the "anchor" variable[4] $\mathbf{z}^*$ is unknown and must be estimated, and 2) it is necessary to ensure that the estimated $\mathbf{z}^*$ lies in the support set of training distribution, as specified by the support constraints in Equation 5.

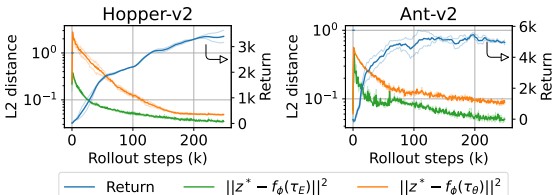

In summary, by comparing $\mathcal{J}_{\mathrm{MI}(f_\phi)}$ and $\mathcal{J}_D$, we can observe that $\mathcal{J}_{\mathrm{MI}(f_\phi)}$ actually encourages expert matching in the embedding space, while $\mathcal{J}_D$ encourages expert matching in the original trajectory space. In the next section, we will see that such an embedding-level expert matching objective naturally lends itself to cross-domain IL settings.

Figure 1: During learning, the distance between $\mathbf{z}^*$ and $f_\phi(\boldsymbol{\tau}_E)$ decreases rapidly (green lines). Meanwhile, as policy $\pi_\theta(\cdot|\cdot, \mathbf{z}^*)$ gets better (blue lines), $f_\phi(\boldsymbol{\tau}_\theta)$ gradually approaches $\mathbf{z}^*$ (red lines).

### 4.3 Practical Implementation

In this section, we describe how we can convert the bi-level IL problem above (Equations 4 and 5) into a feasible online/offline IL objective and discuss some practical implementation details in LfO, offline IL, cross-domain IL, and one-shot IL settings (see more details[5] in Appendix 9.3).

As shown in Algorithm 1 (best viewed in colors), CEIL alternates between solving the bi-level problem with respect to the support constraint (Line 3 for online IL or Line 7 for offline IL), the trajectory self-consistency loss (Line 5), and the optimal embedding inference (Line 6).

To satisfy the support constraint in Equation 5, for online IL (Line 3), we directly roll out the $z^*$-conditioned policy $\pi_\theta(\mathbf{a}|\mathbf{s}, \mathbf{z}^*)$ in the environment; for offline IL (Line 7), we minimize a simple regularization[6] over $\mathbf{z}^*$, bearing a close resemblance to the one used in TD3+BC [23]:

$$\mathcal{R}(\mathbf{z}^*) = \min\left(\|\mathbf{z}^* - f_{\bar{\phi}}(\boldsymbol{\tau}_E)\|^2, \|\mathbf{z}^* - f_{\bar{\phi}}(\boldsymbol{\tau}_\mathcal{D})\|^2\right), \quad \boldsymbol{\tau}_E := \boldsymbol{\tau} \sim \pi_E(\boldsymbol{\tau}), \ \boldsymbol{\tau}_\mathcal{D} := \boldsymbol{\tau} \sim \mathcal{D}(\boldsymbol{\tau}), \quad (7)$$

where we apply a stop-gradient operation to $f_{\bar{\phi}}$. To ensure the optimal embedding inference $(\max_{\mathbf{z}^*} \mathcal{J}_{\mathrm{MI}(f_\phi)} - \mathcal{J}_D)$ retaining the flexibility of seeking $\mathbf{z}^*$ across different instances of $f_\phi$, we jointly update the optimal embedding $\mathbf{z}^*$ and the embedding function $f_\phi$ with

$$\max_{\mathbf{z}^*, f_\phi} \ \mathcal{J}_{\mathrm{MI}(f_\phi)} - \alpha \mathcal{J}_D, \quad (8)$$

where we use $\alpha$ to control the weight on $\mathcal{J}_D$.

**LfO.** In the LfO setting, as expert actions are missing, we apply our expert matching objective only over the observations. Note that even though expert data contains no actions in LfO, we can still

---

[4] The triplet contrastive loss enforces the distance between the anchor and the positive to be smaller than that between the anchor and the negative. Thus, we can view $\mathbf{z}^*$ in $\mathcal{J}_{\mathrm{MI}(f_\phi)}$ as an instance of the anchor.

[5] Our code will be released at `https://github.com/wechto/GeneralizedCEIL`.

[6] In other words, the offline support constraint in Equation 5 is achieved through minimizing $\mathcal{R}(\mathbf{z}^*)$.

leverage a large number of suboptimal actions presented in online/offline $\mathcal{D}(\boldsymbol{\tau})$. Thus, we can learn the contextual policy $\pi_\theta(\mathbf{a}|\mathbf{s}, \mathbf{z})$ using the buffer data in online IL or the offline data in offline IL, much owing to the fact that we do not directly use the plain expert matching objective to update $\pi_\theta$.

**Cross-domain IL.** Cross-domain IL considers the case in which the expert's and learning agent's MDPs are different. Due to the domain shift, the plain idea of $\min \mathcal{J}_D$ may not be a sufficient proxy for the expert matching objective, as there may never exist a trajectory (in the learning MDP) that matches the given expert data. Thus, we can set (the weight of $\mathcal{J}_D$) $\alpha$ to 0.

Further, to make embedding function $f_\phi$ useful for guiding the expert matching in latent space (*i.e.*, $\max \mathcal{J}_{\mathrm{MI}(f_\phi)}$), we encourage $f_\phi$ to capture the task-relevant embeddings and ignore the domain-specific factors. To do so, we generate a set of pseudo-random transitions $\{\boldsymbol{\tau}_{E'}\}$ by independently sampling trajectories from expert data $\{\pi_E(\boldsymbol{\tau}_E)\}$ and adding random noise over these sampled trajectories, *i.e.*, $\boldsymbol{\tau}_{E'} = \boldsymbol{\tau}_E + \text{noise}$. Then, we couple each trajectory $\boldsymbol{\tau}$ in $\{\boldsymbol{\tau}_E\} \cup \{\boldsymbol{\tau}_{E'}\}$ with a label $\mathbf{n} \in \{\mathbf{0}, \mathbf{1}\}$, indicating whether it is noised, and then generate a new set of $\{(\boldsymbol{\tau}, \mathbf{n})\}$, where $\boldsymbol{\tau} \in \{\boldsymbol{\tau}_E\} \cup \{\boldsymbol{\tau}_{E'}\}$ and $\mathbf{n} \in \{\mathbf{0}, \mathbf{1}\}$. Thus, we can set the regularization $\mathcal{R}(f_\phi)$ in Equation 5 to be:

$$\mathcal{R}(f_\phi) = \mathcal{I}(f_\phi(\boldsymbol{\tau}); \mathbf{n}). \tag{9}$$

Intuitively, maximizing $\mathcal{R}(f_\phi)$ encourages embeddings to be domain-agnostic and task-relevant: $f_\phi(\boldsymbol{\tau}_E)$ has high predictive power over expert data ($\mathbf{n} = 0$) and low that over noised data ($\mathbf{n} = \mathbf{1}$).

**One-shot IL.** Benefiting from the separate design of the contextual policy learning and the optimal embedding inference, CEIL also enjoys another advantage — one-shot generalization to new IL tasks. For new IL tasks, given the corresponding expert data $\boldsymbol{\tau}_{\text{new}}$, we can use the learned embedding function $f_\phi$ to generate a corresponding latent embedding $\mathbf{z}_{\text{new}}$. When conditioning on such an embedding, we can directly roll out $\pi_\theta(\mathbf{a}|\mathbf{s}, \mathbf{z}_{\text{new}})$ to recover the one-shot expert behavior.

## 5 Experiments

In this section, we conduct experiments across a variety of IL problem domains: single/cross-domain IL, online/offline IL, and LfD/LfO IL settings. By arranging and combining these IL domains, we obtain 8 IL tasks in all: *S-on-LfD*, *S-on-LfO*, *S-off-LfD*, *S-off-LfO*, *C-on-LfD*, *C-on-LfO*, *C-off-LfD*, and *C-off-LfO*, where S/C denotes single/cross-domain IL, on/off denotes online/offline IL, and LfD/LfO denote learning from demonstrations/observations respectively. Moreover, we also verify the scalability of CEIL on the challenging one-shot IL setting.

Our experiments are conducted in four popular MuJoCo environments: Hopper-v2 (Hop.), HalfCheetah-v2 (Hal.), Walker2d-v2 (Wal.), and Ant-v2 (Ant.). In the single-domain IL setting, we train a SAC policy in each environment and use the learned expert policy to collect expert trajectories (demonstrations/observations). To investigate the cross-domain IL setting, we assume the two domains (learning MDP and the expert-data collecting MDP) have the same state space and action space, while they have different transition dynamics. To achieve this, we modify the torso length of the MuJoCo agents (see details in Appendix 9.2). Then, for each modified agent, we train a separate expert policy and collect expert trajectories. For the offline IL setting, we directly take the reward-free D4RL [22] as the offline dataset, replacing the online rollout experience in the online IL setting.

### 5.1 Evaluation Results

To demonstrate the versatility of the CEIL idea, we collect 20 expert trajectories (demonstrations in LfD or state-only observations in LfO) for each environment and compare CEIL to GAIL [30], AIRL [21], SQIL [61], IQ-Learn [28], ValueDICE [41], GAIfO [66], ORIL [78], DemoDICE [39], and SMODICE [56] (see their implementation details in Appendix 9.4). Note that these baseline methods cannot be applied to all the IL task settings (*S/C-on/off-LfD/LfO*), thus we only provide experimental comparisons with compatible baselines in each IL setting.

**Online IL.** In Figure 2, we provide the return (cumulative rewards) curves of our method and baselines on 4 online IL settings: *S-on-LfD* (*top-left*), *S-on-LfO* (*top-right*), *C-on-LfD* (*bottom-left*), and *C-on-LfO* (*bottom-right*) settings. As can be seen, CEIL quickly achieves expert-level performance in *S-on-LfD*. When extended to *S-on-LfO*, CEIL also yields better sample efficiency compared to baselines. Further, considering the complex cross-domain setting, we can see those baselines SQIL

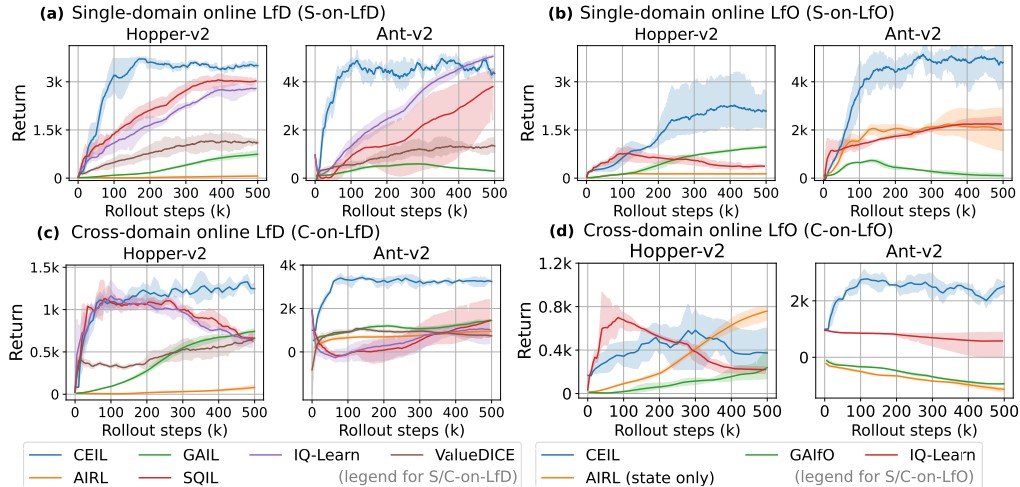

Figure 2: Return curves on 4 online IL settings: (**a**) *S-on-LfD*, (**b**) *S-on-LfO*, (**c**) *C-on-LfD*, and (**d**) *C-on-LfO*, where the shaded area represents a 95% confidence interval over 30 trails. Note that baselines cannot be applied to all the IL task settings, thus we only provide comparisons with compatible baselines (two separate legends).

Table 2: Normalized scores (averaged over 30 trails for each task) on 4 offline IL settings: *S-off-LfD*, *S-off-LfO*, *C-off-LfD*, and *C-off-LfO*. Scores within two points of the maximum score are highlighted

| | Offline IL settings | Hopper-v2 | | | Halfcheetah-v2 | | | Walker2d-v2 | | | Ant-v2 | | | sum |
|---|---|---|---|---|---|---|---|---|---|---|---|---|---|---|
| | | m | mr | me | m | mr | me | m | mr | me | m | mr | me | |
| S-LfD | ORIL (TD3+BC) | 50.9 | 22.1 | 72.7 | 44.7 | 30.2 | 87.5 | 47.1 | 26.7 | 102.6 | 46.5 | 31.4 | 61.9 | 624.3 |
| | SQIL (TD3+BC) | 32.6 | 60.6 | 25.5 | 13.2 | 25.3 | 14.4 | 25.6 | 15.6 | 8.0 | 63.6 | 58.4 | 44.3 | 387.1 |
| | IQ-Learn | 21.3 | 19.9 | 24.9 | 5.0 | 7.5 | 7.5 | 22.3 | 19.6 | 18.5 | 38.4 | 24.3 | 55.3 | 264.5 |
| | ValueDICE | 73.8 | 83.6 | 50.8 | 1.9 | 2.4 | 3.2 | 24.6 | 26.4 | 44.1 | 79.1 | 82.4 | 75.2 | 547.5 |
| | DemoDICE | 54.8 | 32.7 | 65.4 | 42.8 | 37.0 | 55.6 | 68.1 | 39.7 | 95.0 | 85.6 | 69.0 | 108.8 | 754.6 |
| | SMODICE | 56.1 | 28.7 | 68.0 | 42.7 | 37.7 | 66.9 | 66.2 | 40.7 | 58.2 | 87.4 | 69.9 | 113.4 | 735.9 |
| | **CEIL (ours)** | 110.4 | 103.0 | 106.8 | 40.0 | 30.3 | 63.9 | 118.6 | 110.8 | 117.0 | 126.3 | 122.0 | 114.3 | **1163.5** |
| S-LfO | ORIL (TD3+BC) | 43.4 | 25.7 | 73.0 | 44.9 | 2.4 | 81.8 | 58.9 | 16.8 | 78.2 | 33.7 | 29.6 | 67.1 | 555.4 |
| | SMODICE | 54.5 | 26.4 | 73.7 | 42.7 | 37.9 | 66.2 | 60.6 | 38.5 | 70.9 | 85.7 | 68.3 | 116.3 | 741.7 |
| | **CEIL (ours)** | 54.2 | 51.4 | 90.4 | 43.5 | 40.1 | 47.7 | 78.5 | 20.5 | 110.0 | 97.0 | 67.8 | 120.5 | **821.7** |
| C-LfD | ORIL (TD3+BC) | 52.8 | 27.6 | 46.5 | 38.3 | 8.0 | 74.0 | 25.3 | 28.4 | 26.3 | 26.0 | 17.6 | 11.9 | 382.6 |
| | SQIL (TD3+BC) | 34.4 | 19.1 | 11.4 | 19.2 | 25.1 | 19.9 | 15.8 | 16.5 | 8.8 | 21.8 | 23.2 | 21.2 | 236.2 |
| | IQ-Lean | 37.3 | 35.4 | 25.9 | 27.4 | 27.1 | 31.2 | 27.7 | 22.2 | 31.7 | 63.7 | 63.3 | 55.8 | 448.8 |
| | ValueDICE | 22.0 | 18.3 | 18.9 | 14.0 | 11.7 | 8.7 | 11.5 | 10.0 | 8.6 | 24.1 | 24.1 | 19.2 | 188.4 |
| | DemoDICE | 52.9 | 15.2 | 77.2 | 42.8 | 38.9 | 53.8 | 58.4 | 26.4 | 77.8 | 87.8 | 69.3 | 114.9 | 715.6 |
| | SMODICE | 55.4 | 21.4 | 71.2 | 42.7 | 38.0 | 64.6 | 68.4 | 34.2 | 80.4 | 87.4 | 70.4 | 115.7 | 749.7 |
| | **CEIL (ours)** | 58.4 | 39.8 | 81.6 | 42.6 | 38.3 | 46.6 | 76.5 | 21.1 | 81.1 | 91.6 | 88.0 | 115.3 | **780.9** |
| C-LfO | ORIL (TD3+BC) | 55.5 | 18.2 | 55.5 | 40.6 | 2.9 | 73.0 | 26.9 | 19.4 | 22.7 | 11.2 | 21.3 | 10.8 | 358.0 |
| | SMODICE | 53.7 | 18.3 | 64.2 | 42.6 | 38.0 | 63.0 | 68.9 | 37.5 | 60.7 | 87.5 | 75.1 | 115.0 | **724.4** |
| | **CEIL (ours)** | 44.7 | 44.2 | 48.2 | 42.4 | 36.5 | 46.9 | 76.2 | 31.7 | 77.0 | 95.9 | 71.0 | 112.7 | **727.3** |

and IQ-Learn (in *C-on-LfD* and *C-on-LfO*) suffer from the domain mismatch, leading to performance degradation at late stages of training, while CEIL can still achieve robust performance.

**Offline IL.** Next, we evaluate CEIL on the other 4 offline IL settings: *S-off-LfD*, *S-off-LfO*, *C-off-LfD*, and *C-off-LfO*. In Table 2, we provide the normalized return of our method and baseline methods on reward-free D4RL [22] medium (m), medium-replay (mr), and medium-expert (me) datasets. We can

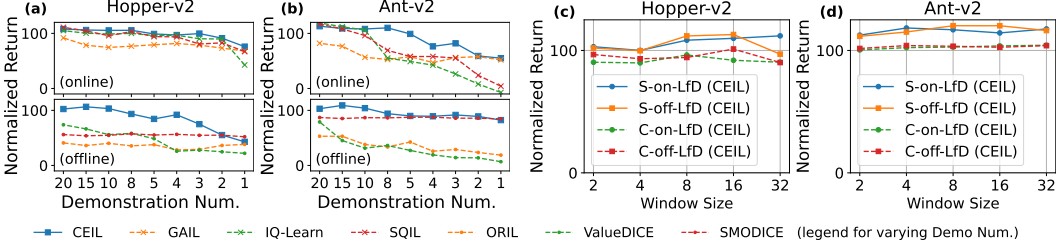

Figure 3: Ablating (**a**, **b**) the number of expert demonstrations and (**c**, **d**) the trajectory window size.

observe that CEIL achieves a significant improvement over the baseline methods in both *S-off-LfD* and *S-off-LfO* settings. Compared to the state-of-the-art offline baselines, CEIL also shows competitive results on the challenging cross-domain offline IL settings (*C-off-LfD* and *C-off-LfO*).

**One-shot IL.** Then, we explore CEIL on the one-shot IL tasks, where we expect CEIL can adapt its behavior to new IL tasks given only one trajectory for each task (mismatched MDP, see Appendix 9.2).

We first pre-train an embedding function and a contextual policy in the training domain (online/offline IL), then infer a new contextual variable and evaluate it on the new task. To facilitate comparison to baselines, we similarly pre-train a policy network (using baselines) and run BC on top of the pre-trained policy by using the provided demonstration. Consequently, such a baseline+BC procedure cannot be applied to the (one-shot) LfO tasks. The results in Table 3 show that baseline+BC struggles to transfer their expertise to new tasks. Benefiting from the hindsight framework, CEIL shows better one-shot transfer learning performance on 7 out of 8 one-shot LfD tasks and retains higher scalability and generality for both one-shot LfD and LfO IL tasks.

| One-shot IL | | Hop. | Hal. | Wal. | Ant. |
|---|---|---|---|---|---|
| Online | SQIL | 16.8 | 1.1 | 3.5 | 4.2 |
| | IQ-Learn | 4.6 | 0.2 | 1.7 | 7.5 |
| | **CEIL** (LfD) | 29.9 | 2.5 | 31.7 | 20.5 |
| | **CEIL** (LfO) | 17.8 | 3.2 | 5.6 | 29.7 |
| Offline | ORIL | 14.7 | 0.2 | 6.9 | 17.4 |
| | SQIL | 7.4 | 0.8 | 4.6 | 12.5 |
| | IQ-Learn | 18.8 | 1.2 | 4.0 | 19.3 |
| | DemoDICE | 76.5 | -0.5 | -0.1 | 19.5 |
| | SMODICE | 78.0 | 1.1 | 8.1 | 24.6 |
| | **CEIL** (LfD) | 85.6 | 5.6 | 67.1 | 24.3 |
| | **CEIL** (LfO) | 72.2 | 5.1 | 70.0 | 19.4 |

Table 3: Normalized results on one-shot IL, where CEIL shows prominent transferability.

## 5.2 Analysis of CEIL

**Hybrid IL settings.** In real-world, many IL tasks do not correspond to one specific IL setting, and instead consist of a hybrid of several IL settings, each of which passes a portion of task-relevant information to the IL agent. For example, we can provide the agent with both demonstrations and state-only observations and, in some cases, cross-domain demonstrations (S-LfD+S-LfO+C-LfD).

| Hybrid offline IL settings | Hop. | Hal. | Wal. | Ant. |
|---|---|---|---|---|
| S-LfD | 29.4 | 69.9 | 42.8 | 84.9 |
| S-LfD + S-LfO | 30.4 | 68.6 | 42.3 | 91.6 |
| S-LfD + S-LfO + C-LfD | 30.7 | 71.7 | 42.9 | 89.2 |
| S-LfD + S-LfO + C-LfD + C-LfO | 58.6 | 79.6 | 43.7 | 98.0 |

Table 4: The normalized results of CEIL, showing that CEIL can consistently digest useful (task-relevant) information and boost its performance, even under a hybrid of offline IL settings.

To examine the versatility of CEIL, we collect a separate expert trajectory for each of the four offline IL settings, and study CEIL's performance under hybrid IL settings. As shown in Table 4, we can see that by adding new expert behaviors on top of LfD, even when carrying relatively less supervision (*e.g.*, actions are absent in LfO), CEIL can still improve the performance.

**Varying the number of demonstrations.** In Figure 3 (a, b), we study the effect of the number of expert demonstrations on CEIL's performance. Empirically, we reduce the number of training demonstrations from 20 to 1, and report the normalized returns at 1M training steps. We can observe that across both online and offline (D4RL *-medium) IL settings, CEIL shows more robust performance with respect to different numbers of demonstrations compared to baseline methods.

**Varying the window size of trajectory.** Next we assess the effect of the trajectory window size (*i.e.*, the length of trajectory $\tau$ used for the embedding function $f_\phi$ in Equation 3). In Figure 3 (b, c), we ablate the number of the window size in 4 LfD IL instantiations. We can see that across a range of window sizes, CEIL remains stable and achieves expert-level performance.

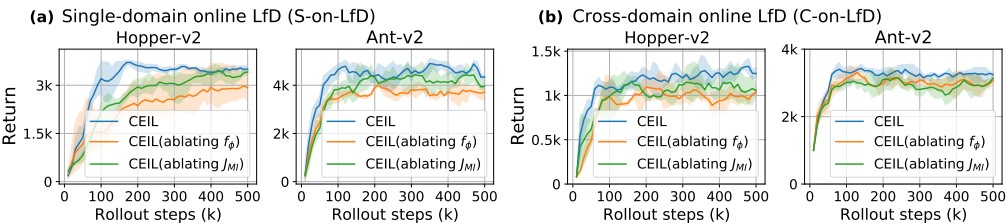

Figure 4: **Ablation studies on the optimization of $f_\phi$ (*ablating* $f_\phi$) and the objective of $\mathcal{J}_{\mathbf{MI}}$ (*ablating* $\mathcal{J}_{\mathbf{MI}}$),** where the shaded area represents 95% CIs over 5 trails. See ablation results for offline IL tasks in Table 5.

Table 5: **Ablation studies on the optimization of $f_\phi$ (*ablating $f_\phi$*) and the objective of $\mathcal{J}_{\mathbf{MI}}$ (*ablating $\mathcal{J}_{\mathbf{MI}}$*),** where scores (averaged over 5 trails for each task) within two points of the maximum score are highlighted.

| | Offline IL settings | Hopper-v2 | | | HalfCheetah-v2 | | | Walker2d-v2 | | | Ant-v2 | | | sum |
|---|---|---|---|---|---|---|---|---|---|---|---|---|---|---|
| | | m | mr | me | m | mr | me | m | mr | me | m | mr | me | |
| S-LfD | CEIL (*ablating $f_\phi$*) | 97.9 | 92.5 | 99.3 | 41.3 | 30.3 | 66.7 | 103.6 | 88.1 | 114.4 | 97.6 | 98.4 | 100.7 | 1030.8 |
| | CEIL (*ablating $\mathcal{J}_{MI}$*) | 83.2 | 89.0 | 98.7 | 27.1 | 28.3 | 53.5 | 107.4 | 68.0 | 75.6 | 116.9 | 97.8 | 105.9 | 951.4 |
| | CEIL | 110.4 | 103.0 | 106.8 | 40.0 | 30.3 | 63.9 | 118.6 | 110.8 | 117.0 | 126.3 | 122.0 | 114.3 | **1163.5** |
| S-LfO | CEIL (*ablating $f_\phi$*) | 51.5 | 41.1 | 83.3 | 43.8 | 40.1 | 63.7 | 76.3 | 20.3 | 103.0 | 78.0 | 52.5 | 105.5 | 759.2 |
| | CEIL (*ablating $\mathcal{J}_{MI}$*) | 54.3 | 44.9 | 84.7 | 42.2 | 39.9 | 51.6 | 77.4 | 22.7 | 94.0 | 92.1 | 67.9 | 118.4 | 792.0 |
| | CEIL | 54.2 | 51.4 | 90.4 | 43.5 | 40.1 | 47.7 | 78.5 | 20.5 | 110.0 | 97.0 | 67.8 | 120.5 | **821.7** |

**Ablation studies on the optimization of $f_\phi$ and the objective of $\mathcal{J}_{\mathbf{MI}}$.** In Figure 4 and Table 5, we carried out ablation experiments on the loss of $f_\phi$ and $\mathcal{J}_{\mathbf{MI}}$ in both online IL and offline IL settings. We can see that ablating the $f_\phi$ loss (optimizing with Equation 5) does degrade the performance in both online and offline IL tasks, demonstrating the effectiveness of optimizing with Equation 8. Intuitively, Equation 8 encourages the embedding function to be task-relevant, and thus we use the expert matching loss to update $f_\phi$. We can also see that ablating $\mathcal{J}_{\mathbf{MI}}$ does lead to degraded performance, further verifying the effectiveness of our expert matching objective in the latent space.

## 6 Conclusion

In this paper, we present CEIL, a novel and general Imitation Learning framework applicable to a wide range of IL settings, including *C/S-on/off-LfD/LfO* and few-shot IL settings. This is achieved by explicitly decoupling the imitation policy into 1) a contextual policy, learned with the self-supervised hindsight information matching objective, and 2) a latent variable, inferred by performing the IL expert matching objective. Compared to prior baselines, our results show that CEIL is more sample-efficient in most of the online IL tasks and achieves better or competitive performances in offline tasks.

**Limitations and future work.** Our primary aim behind this work is to develop a simple and scalable IL method. We believe that CEIL makes an important step in that direction. Admittedly, we also find some limitations of CEIL: 1) Offline results generally outperform online results, especially in the LfO setting. The main reason is that CEIL lacks explicit exploration bounds, thus future work could explore the exploration ability of online CEIL. 2) The trajectory self-consistency cannot be applied to cross-embodiment agents once the two embodiments/domains have different state spaces or action spaces. Considering such a cross-embodiment setting, a typical approach is to serialize state/action from different modalities into a flat sequence of tokens. We also remark that CEIL is compatible with such a tokenization approach, and thus suitable for IL tasks with different action/state spaces. Thus, we encourage the future exploration of generalized IL methods across different embodiments.

## Acknowledgments and Disclosure of Funding

We sincerely thank the anonymous reviewers for their insightful suggestions. This work was supported by the National Science and Technology Innovation 2030 - Major Project (Grant No. 2022ZD0208800), and NSFC General Program (Grant No. 62176215).

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

# Appendix

## 7 Additional Derivation

(*Repeat from the main paper.*) To gain more insight into Equation 4 that captures the quality of IL (the degree of similarity to the expert data), we define $D(\cdot, \cdot)$ as the sum of reverse KL and forward KL divergence, *i.e.*, $D(q, p) = D_{\mathrm{KL}}(q\|p) + D_{\mathrm{KL}}(p\|q)$, and derive an alternative form for Equation 4:

$$\arg \min_{\mathbf{z}^*} D(\pi_\theta(\boldsymbol{\tau}|\mathbf{z}^*), \pi_E(\boldsymbol{\tau})) = \arg \max_{\mathbf{z}^*} \underbrace{\mathcal{I}(\mathbf{z}^*; \boldsymbol{\tau}_E) - \mathcal{I}(\mathbf{z}^*; \boldsymbol{\tau}_\theta)}_{\mathcal{J}_{\mathrm{MI}}} - \underbrace{D_{\mathrm{KL}}(\pi_\theta(\boldsymbol{\tau}), \pi_E(\boldsymbol{\tau}))}_{\mathcal{J}_D},$$

where $\mathcal{I}(\mathbf{x}; \mathbf{y})$ denotes the mutual information (MI) between $\mathbf{x}$ and $\mathbf{y}$, which measures the predictive power of $\mathbf{y}$ on $\mathbf{x}$ (or vice-versa), the latent variables are defined as $\boldsymbol{\tau}_E := \boldsymbol{\tau} \sim \pi_E(\boldsymbol{\tau})$, $\boldsymbol{\tau}_\theta := \boldsymbol{\tau} \sim p(\mathbf{z}^*)\pi_\theta(\boldsymbol{\tau}|\mathbf{z}^*)$, and $\pi_\theta(\boldsymbol{\tau}) = \mathbb{E}_{\mathbf{z}^*}[\pi_\theta(\boldsymbol{\tau}|\mathbf{z}^*)]$.

Below is our derivation:

$$\min_{\mathbf{z}^*} D(\pi_\theta(\boldsymbol{\tau}|\mathbf{z}^*), \pi_E(\boldsymbol{\tau}))$$
$$= \min_{\mathbf{z}^*} \mathbb{E}_{p(\mathbf{z}^*)} [D_{\mathrm{KL}}(\pi_\theta(\boldsymbol{\tau}|\mathbf{z}^*)\|\pi_E(\boldsymbol{\tau})) + D_{\mathrm{KL}}(\pi_E(\boldsymbol{\tau})\|\pi_\theta(\boldsymbol{\tau}|\mathbf{z}^*))]$$
$$= \min_{\mathbf{z}^*} \mathbb{E}_{p(\mathbf{z}^*)\pi_\theta(\boldsymbol{\tau}|\mathbf{z}^*)} [\log \pi_\theta(\boldsymbol{\tau}|\mathbf{z}^*) - \log \pi_E(\boldsymbol{\tau})]$$
$$\qquad + \mathbb{E}_{p(\mathbf{z}^*)\pi_E(\boldsymbol{\tau})} [\log \pi_E(\boldsymbol{\tau}) - \log \pi_\theta(\boldsymbol{\tau}|\mathbf{z}^*)]$$
$$= \min_{\mathbf{z}^*} \mathbb{E}_{p(\mathbf{z}^*)\pi_\theta(\boldsymbol{\tau}|\mathbf{z}^*)} \left[\log \frac{p(\mathbf{z}^*|\boldsymbol{\tau})\pi_\theta(\boldsymbol{\tau})}{p(\mathbf{z}^*)} - \log \pi_E(\boldsymbol{\tau})\right]$$
$$\qquad + \mathbb{E}_{p(\mathbf{z}^*)\pi_E(\boldsymbol{\tau})} \left[\log \pi_E(\boldsymbol{\tau}) - \log \frac{p(\mathbf{z}^*|\boldsymbol{\tau})\pi_\theta(\boldsymbol{\tau})}{p(\mathbf{z}^*)}\right]$$
$$= \min_{\mathbf{z}^*} \mathbb{E}_{p(\mathbf{z}^*)\pi_\theta(\boldsymbol{\tau}|\mathbf{z}^*)} \left[\log \frac{p(\mathbf{z}^*|\boldsymbol{\tau})}{p(\mathbf{z}^*)} + \log \frac{\pi_\theta(\boldsymbol{\tau})}{\pi_E(\boldsymbol{\tau})}\right] - \mathbb{E}_{p(\mathbf{z}^*)\pi_E(\boldsymbol{\tau})} \left[\log \frac{p(\mathbf{z}^*|\boldsymbol{\tau})}{p(\mathbf{z}^*)} + \log \frac{\pi_\theta(\boldsymbol{\tau})}{\pi_E(\boldsymbol{\tau})}\right]$$
$$= \max_{\mathbf{z}^*} \mathcal{I}(\mathbf{z}^*; \boldsymbol{\tau}_E) - \mathcal{I}(\mathbf{z}^*; \boldsymbol{\tau}_\theta) - D(\pi_\theta(\boldsymbol{\tau}), \pi_E(\boldsymbol{\tau})),$$

where $\boldsymbol{\tau}_E := \boldsymbol{\tau} \sim \pi_E(\boldsymbol{\tau})$, $\boldsymbol{\tau}_\theta := \boldsymbol{\tau} \sim p(\mathbf{z}^*)\pi_\theta(\boldsymbol{\tau}|\mathbf{z}^*)$.

## 8 More Comparisons and Ablation Studies

### 8.1 Offline Comparison on D4RL Expert Domain Dataset

In Table 6, we provide the normalized return of our method and baseline methods on the reward-free D4RL [22] expert dataset. Consistently, we can observe that CEIL achieves a significant improvement over the baseline methods in both *S-off-LfD* and *S-off-LfO* settings. Compared to the state-of-the-art offline IL baselines, CEIL also shows competitive results on the challenging cross-domain offline IL settings (*C-off-LfD* and *C-off-LfO*).

### 8.2 Generalizability on Cross-domain Offline IL Settings

In the standard cross-domain IL setting, the goal is to extract expert-relevant information from the mismatched expert demonstrations/observations (expert domain) and to mimic such expert behaviors in the training environment (training domain). Thus, we validate the performance of the learned policy in the training environment (*i.e.*, the environment where the offline data was collected). Here, we also study the generalizability of the learned policy by evaluating the learned policy in the expert environment (*i.e.*, the environment where the mismatched expert data was collected). We provide the normalized scores (*evaluated in the expert domain*) in Table 7. We can find that across a range of cross-domain offline IL tasks, CEIL consistently demonstrates better (zero-shot) generalizability compared to baselines.

### 8.3 Ablating the Cross-domain Regularization

We now conduct ablation studies to evaluate the importance of cross-domain regularization in Equation 9 (in the main paper). In Figure 5, we provide the performance improvement when we

Table 6: Normalized scores (averaged over 30 trails for each task) on D4RL expert dataset. Scores within two points of the maximum score are highlighted. hop: Hopper-v2. hal: HalfCheetah-v2. wal: Walker2d-v2. ant: Ant-v2.

| | | hop expert | hal expert | wal expert | ant expert | sum |
|---|---|---|---|---|---|---|
| *S-off-LfD* | ORIL (TD3+BC) | 97.5 | 91.8 | 14.5 | 76.8 | 280.6 |
| | SQIL (TD3+BC) | 25.5 | 14.4 | 8.0 | 44.3 | 92.1 |
| | IQ-Learn | 37.3 | 9.9 | 46.6 | 85.9 | 179.7 |
| | ValueDICE | 65.6 | 2.9 | 28.2 | 90.5 | 187.1 |
| | DemoDICE | 107.3 | 87.1 | 104.8 | 114.2 | 413.3 |
| | SMODICE | 111.0 | 93.5 | 108.2 | 122.0 | **434.7** |
| | CEIL | 106.0 | 96.0 | 115.6 | 117.8 | **435.4** |
| *S-off-LfO* | ORIL (TD3+BC) | 64.2 | 92.1 | 12.2 | 44.3 | 212.8 |
| | SMODICE | 111.3 | 93.7 | 108.0 | 122.0 | **435.0** |
| | CEIL | 103.3 | 96.8 | 110.0 | 126.4 | **436.5** |
| *C-off-LfD* | ORIL (TD3+BC) | 24.4 | 78.3 | 29.3 | 32.1 | 164.1 |
| | SQIL (TD3+BC) | 12.2 | 19.9 | 8.8 | 21.2 | 62.0 |
| | IQ-Learn | 25.9 | 31.2 | 31.7 | 55.8 | 144.6 |
| | ValueDICE | 18.6 | 9.8 | 8.3 | 22.3 | 59.0 |
| | DemoDICE | 111.5 | 88.7 | 107.9 | 122.5 | 430.6 |
| | SMODICE | 111.1 | 93.8 | 108.2 | 120.9 | **434.0** |
| | CEIL | 105.8 | 97.1 | 108.6 | 112.2 | 423.7 |
| *C-off-LfO* | ORIL (TD3+BC) | 22.5 | 76.6 | 11.2 | 28.2 | 138.6 |
| | SMODICE | 111.2 | 93.7 | 108.1 | 117.7 | 430.7 |
| | CEIL | 113.0 | 90.1 | 108.7 | 125.2 | **437.0** |

Table 7: Normalized scores (*evaluated on the expert dataset* over 30 trails for each task) on 2 cross-domain offline IL settings: *C-off-LfD* and *C-off-LfO*. Scores within two points of the maximum score are highlighted. m: medium. mr: medium-replay. me: medium-expert. e: expert.

| | | Hopper-v2 | | | | HalfCheetah-v2 | | | | sum |
|---|---|---|---|---|---|---|---|---|---|---|
| | | m | mr | me | e | m | mr | me | e | |
| C-off-LfD | ORIL (TD3+BC) | 74.7 | 16.7 | 45.0 | 21.4 | 2.2 | 0.8 | -0.3 | -2.2 | 158.3 |
| | SQIL (TD3+BC) | 33.6 | 21.6 | 14.5 | 14.5 | 18.2 | 7.5 | 20.9 | 20.9 | 151.8 |
| | IQ-Learn | 11.8 | 9.7 | 17.1 | 17.1 | 7.7 | 7.8 | 9.5 | 9.5 | 90.2 |
| | ValueDICE | 49.5 | 24.2 | 55.7 | 49.3 | 32.2 | 32.9 | 38.7 | 28.7 | 311.2 |
| | DemoDICE | 83.2 | 31.5 | 81.6 | 28.5 | 0.9 | -1.1 | -1.7 | -2.4 | 220.6 |
| | SMODICE | 80.1 | 26.1 | 78.0 | 54.3 | 2.8 | -1.0 | 1.0 | -2.3 | 239.1 |
| | CEIL | 87.4 | 74.3 | 81.2 | 82.4 | 44.0 | 30.4 | 25.0 | 17.1 | **441.9** |
| *C-off-Lf*O | ORIL (TD3+BC) | 62.3 | 18.7 | 57.0 | 28.2 | 0.2 | 1.1 | -0.3 | -2.3 | 165.0 |
| | SMODICE | 77.6 | 22.5 | 80.2 | 71.0 | 2.0 | -0.9 | 0.8 | -2.3 | 250.9 |
| | CEIL | 56.4 | 58.6 | 56.7 | 65.2 | 5.5 | 36.5 | 5.0 | 5.0 | **288.7** |

| | | Walker2d-v2 | | | | Ant-v2 | | | | sum |
|---|---|---|---|---|---|---|---|---|---|---|
| | | m | mr | me | e | m | mr | me | e | |
| *C-off-LfD* | ORIL (TD3+BC) | 22.0 | 24.5 | 23.9 | 33.1 | 16.0 | 18.6 | 2.5 | 0.4 | 141.0 |
| | SQIL (TD3+BC) | 32.4 | 14.9 | 10.3 | 10.3 | 71.4 | 63.6 | 60.1 | 60.1 | 323.1 |
| | IQ-Learn | 8.4 | 5.0 | 10.2 | 10.2 | 19.4 | 18.4 | 16.1 | 16.1 | 103.8 |
| | ValueDICE | 31.7 | 21.9 | 22.9 | 27.7 | 70.5 | 68.5 | 69.3 | 68.5 | 380.9 |
| | DemoDICE | 12.8 | 31.5 | 12.9 | 86.9 | 15.7 | 24.2 | 2.3 | 1.4 | 187.7 |
| | SMODICE | 43.6 | 16.1 | 62.0 | 85.3 | 23.7 | 22.9 | 2.3 | -5.9 | 249.9 |
| | CEIL | 102.8 | 94.8 | 101.9 | 100.7 | 82.0 | 77.0 | 76.4 | 79.8 | **715.3** |
| *C-off-LfO* | ORIL (TD3+BC) | 22.4 | 15.2 | 17.8 | 12.6 | 13.6 | 20.7 | 5.5 | -6.2 | 101.6 |
| | SMODICE | 42.4 | 17.0 | 55.5 | 88.7 | 15.7 | 22.6 | 2.5 | -6.3 | 238.1 |
| | CEIL | 67.9 | 12.0 | 68.4 | 50.8 | 31.7 | 57.0 | 18.0 | -1.9 | **304.0** |

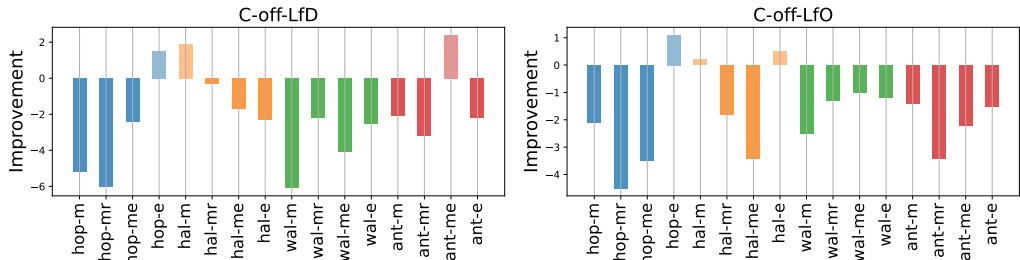

Figure 5: Normalized performance improvement (left: *C-off-LfD*, right: *C-off-LfO*) when we ablate the cross-domain regularization (Equation 9 in the main paper) in cross-domain IL settings. We can observe the general trend (in 26 out of 32 tasks) that ablating the cross-domain regularization causes negative performance improvement. hop: Hopper-v2. hal: HalfCheetah-v2. wal: Walker2d-v2. ant: Ant-v2. m: medium. me: medium-expert. mr: medium-replay. e: expert.

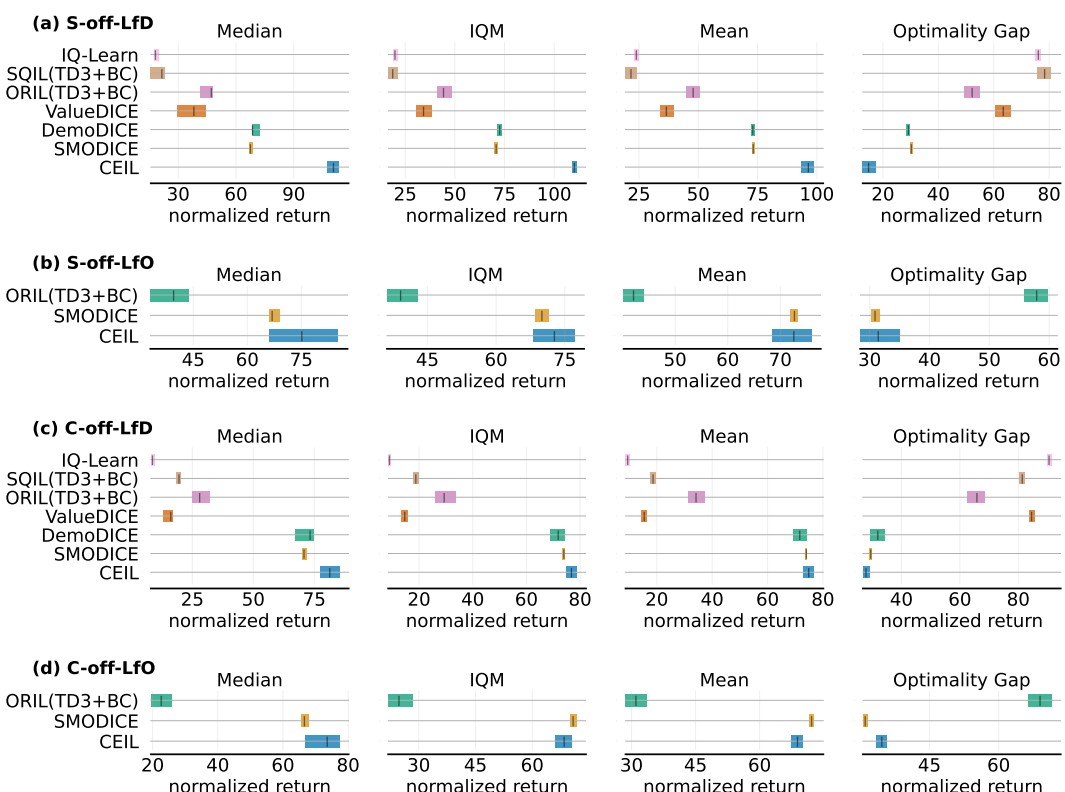

Figure 6: Aggregate median, IQM, mean, and optimality gap over 16 offline IL tasks. Higher median, higher IQM, and higher mean and lower optimality gap are better. The shaded bar shows 95% stratified bootstrap confidence intervals. We can see that CEIL achieves consistently better performance across a wide range of offline IL settings.

ablate the cross-domain regularization in two cross-domain offline IL tasks (*C-off-LfD* and *C-off-LfO*). We can find that in 26 out of 32 cross-domain tasks, ablating the regularization can cause performance to decrease (negative performance improvement), thus verifying the benefits of encouraging task-relevant embeddings.

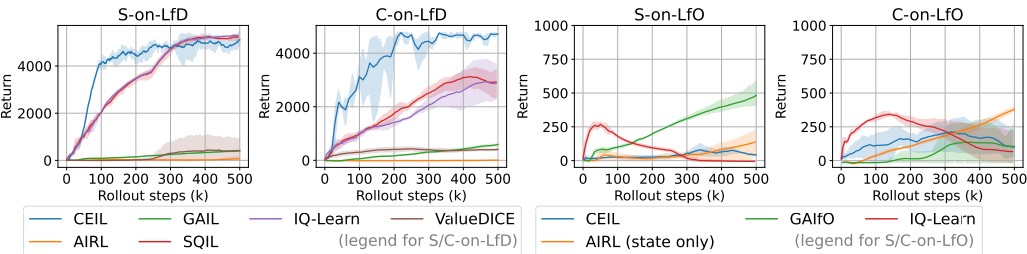

Figure 7: Return curves in Walker2d-v2 (from left to right: *S-on-LfD*, *C-on-LfD*, *S-on-LfO*, and *C-on-LfO*), where the shaded area represents a 95% confidence interval over 30 trails. We can see that CEIL consistently achieves expert-level performance in LfD (*S-on-LfD* and *C-on-LfD*) tasks. Due to the lack of explicit exploration in online LfO settings, CEIL exhibits drastic performance degradation (in *S-on-LfO* and *C-on-LfO*) under the same environmental interaction steps.

## 8.4 Aggregate Results

According to Agarwal et al. [1], we report the aggregate statistics (for 16 offline IL tasks) in Figure 6. We can find that CEIL provides competitive performance consistently across a range of offline IL settings (*S-off-LfD*, *S-off-LfO*, *C-off-LfD*, and *C-off-LfO*) and outperforms prior offline baselines.

Table 8: Normalized scores (averaged over 30 trails for each task) when we vary the number of the expert demonstrations (#5, #10, #15, and #20). Scores within two points of the maximum score are highlighted

| Offline IL settings | | Hopper-v2 | | | Halfcheetah-v2 | | | Walker2d-v2 | | | Ant-v2 | | | sum |
|---|---|---|---|---|---|---|---|---|---|---|---|---|---|---|
| | | m | mr | me | m | mr | me | m | mr | me | m | mr | me | |
| S-off-LfD #5 | ORIL (TD3+BC) | 42.1 | 26.7 | 51.2 | 45.1 | 2.7 | 79.6 | 44.1 | 22.9 | 38.3 | 25.6 | 24.5 | 6.0 | 408.8 |
| | SQIL (TD3+BC) | 45.2 | 27.4 | 5.9 | 14.5 | 15.7 | 11.8 | 12.2 | 7.2 | 13.6 | 20.6 | 23.6 | -5.7 | 192.0 |
| | IQ-Learn | 17.2 | 15.4 | 21.7 | 6.4 | 4.8 | 6.2 | 13.1 | 10.6 | 5.1 | 22.8 | 27.2 | 18.7 | 169.2 |
| | ValueDICE | 59.8 | 80.1 | 72.6 | 2.0 | 0.9 | 1.2 | 2.8 | 0.0 | 7.4 | 27.3 | 32.7 | 30.2 | 316.9 |
| | DemoDICE | 50.2 | 26.5 | 63.7 | 41.9 | 38.7 | 59.5 | 66.3 | 38.8 | 101.6 | 82.8 | 68.8 | 112.4 | 751.2 |
| | SMODICE | 54.1 | 34.9 | 64.7 | 42.6 | 38.4 | 63.8 | 62.2 | 40.6 | 55.4 | 86.0 | 69.7 | 112.4 | 724.7 |
| | CEIL | 94.5 | 45.1 | 80.8 | 45.1 | 43.3 | 33.9 | 103.1 | 81.1 | 99.4 | 99.8 | 101.4 | 85.0 | **912.5** |
| S-off-LfD #10 | ORIL (TD3+BC) | 42.0 | 21.6 | 53.4 | 45.0 | 2.1 | 82.1 | 44.1 | 27.4 | 80.4 | 47.3 | 24.0 | 44.9 | 514.1 |
| | SQIL (TD3+BC) | 50.0 | 34.2 | 7.4 | 8.8 | 10.9 | 8.2 | 20.0 | 15.2 | 9.7 | 35.3 | 36.2 | 11.9 | 247.6 |
| | IQ-Learn | 11.3 | 18.6 | 20.1 | 4.1 | 6.5 | 6.6 | 18.3 | 12.8 | 12.2 | 30.7 | 53.9 | 23.7 | 218.7 |
| | ValueDICE | 56.0 | 64.1 | 54.2 | -0.2 | 2.6 | 2.4 | 4.7 | 4.0 | 0.9 | 31.4 | 72.3 | 49.5 | 341.8 |
| | DemoDICE | 53.6 | 25.8 | 64.9 | 42.1 | 36.9 | 60.6 | 64.7 | 36.1 | 100.2 | 87.4 | 67.1 | 114.3 | 753.5 |
| | SMODICE | 55.6 | 30.3 | 66.6 | 42.6 | 38.0 | 66.0 | 64.5 | 44.6 | 53.8 | 86.9 | 69.5 | 113.4 | 731.8 |
| | CEIL | 113.2 | 53.0 | 96.3 | 64.0 | 43.6 | 44.0 | 120.4 | 82.3 | 104.2 | 119.3 | 70.0 | 90.1 | **1000.4** |
| S-off-LfD #15 | ORIL (TD3+BC) | 38.9 | 22.3 | 46.8 | 44.7 | 1.9 | 83.8 | 37.9 | 4.2 | 69.9 | 59.4 | 22.3 | 12.4 | 444.6 |
| | SQIL (TD3+BC) | 42.8 | 44.4 | 5.2 | 6.8 | 17.1 | 9.1 | 16.9 | 13.5 | 6.9 | 21.2 | 17.2 | 12.6 | 213.6 |
| | IQ-Learn | 14.6 | 8.2 | 29.3 | 4.0 | 3.4 | 5.1 | 7.3 | 14.5 | 11.4 | 54.2 | 15.2 | 61.6 | 228.6 |
| | ValueDICE | 66.3 | 58.3 | 53.6 | 2.3 | 2.3 | 1.2 | 5.2 | -0.1 | 17.0 | 45.2 | 72.0 | 74.3 | 397.8 |
| | DemoDICE | 52.2 | 29.6 | 67.3 | 41.9 | 37.6 | 58.1 | 66.4 | 42.9 | 103.5 | 86.6 | 68.3 | 114.3 | 768.7 |
| | SMODICE | 55.9 | 25.7 | 72.7 | 42.5 | 37.6 | 66.4 | 67.0 | 43.2 | 55.1 | 86.7 | 69.7 | 118.2 | 740.6 |
| | CEIL | 116.4 | 56.7 | 103.7 | 80.4 | 43.0 | 43.8 | 120.3 | 84.8 | 103.8 | 126.8 | 87.0 | 90.6 | **1057.3** |
| S-off-LfD #20 | ORIL (TD3+BC) | 50.9 | 22.1 | 72.7 | 44.7 | 30.2 | 87.5 | 47.1 | 26.7 | 102.6 | 46.5 | 31.4 | 61.9 | 624.3 |
| | SQIL (TD3+BC) | 32.6 | 60.6 | 25.5 | 13.2 | 25.3 | 14.4 | 25.6 | 15.6 | 8.0 | 63.6 | 58.4 | 44.3 | 387.1 |
| | IQ-Learn | 21.3 | 19.9 | 24.9 | 5.0 | 7.5 | 7.5 | 22.3 | 19.6 | 18.5 | 38.4 | 24.3 | 55.3 | 264.5 |
| | ValueDICE | 73.8 | 83.6 | 50.8 | 1.9 | 2.4 | 3.2 | 24.6 | 26.4 | 44.1 | 79.1 | 82.4 | 75.2 | 547.5 |
| | DemoDICE | 54.8 | 32.7 | 65.4 | 42.8 | 37.0 | 55.6 | 68.1 | 39.7 | 95.0 | 85.6 | 69.0 | 108.8 | 754.6 |
| | SMODICE | 56.1 | 28.7 | 68.0 | 42.7 | 37.7 | 66.9 | 66.2 | 40.7 | 58.2 | 87.4 | 69.9 | 113.4 | 735.9 |
| | **CEIL (ours)** | 110.4 | 103.0 | 106.8 | 40.0 | 30.3 | 63.9 | 118.6 | 110.8 | 117.0 | 126.3 | 122.0 | 114.3 | **1163.5** |

## 8.5 Varying the Number of Expert Trajectories

As a complement to the experimental results in the main paper, we continue to compare the performance of CEIL and baselines on more tasks when we vary the number of expert trajectories. Considering offline IL settings, we provide the results in Table 8 for the number of expert trajectories of 5, 10, 15, and 20 respectively. We can find that when varying the number of expert behaviors, CEIL can still obtain higher scores compared to baselines, which is consistent with the findings in Figure 3 in the main paper.

## 8.6 Limitation (Failure Modes in Online LfO Setting)

Meanwhile, we find that in the online LfO settings, CEIL's performance deteriorates severely on a few tasks, as shown in Figure 7 (Walker2d). In LfD (either on single-domain or on cross-domain IL) settings, CEIL can consistently achieve expert-level performance, but when migrating to LfO settings, CEIL suffers collapsing performance under the same number of environmental interactions. We believe that this is due to the lack of expert actions in LfO settings, which causes the agent to stay in the collapsed state region and therefore deteriorates performance. Thus, we believe a rich direction for future research is to explore the online exploration ability.

# 9 Implementation Details

## 9.1 Imitation Learning Tasks

In our paper, we conduct experiments across a variety of IL problem domains: single/cross-domain IL, online/offline IL, and LfD/LfO IL settings. By arranging and combining these IL domains, we obtain 8 IL tasks in all: *S-on-LfD*, *S-on-LfO*, *S-off-LfD*, *S-off-LfO*, *C-on-LfD*, *C-on-LfO*, *C-off-LfD*, and *C-off-LfO*, where S/C denotes single/cross-domain IL, on/off denotes online/offline IL, and LfD/LfO denote learning from demonstrations/observations respectively.

**S-on-LfD.** We have access to a limited number of expert demonstrations and an online interactive training environment. The goal of *S-on-LfD* is to learn an optimal policy that mimics the provided demonstrations in the training environment.

**S-on-LfO.** We have access to a limited number of expert observations (state-only demonstrations) and an online interactive training environment. The goal of *S-on-LfO* is to learn an optimal policy that mimics the provided observations in the training environment.

**S-off-LfD.** We have access to a limited number of expert demonstrations and a large amount of pre-collected offline (reward-free) data. The goal of *S-off-LfD* is to learn an optimal policy that mimics the provided demonstrations in the environment in which the offline data was collected. Note that here *the environment* that was used to collect the expert demonstrations and *the environment* that was used to collect the offline data are the same environment.

**S-off-LfO.** We have access to a limited number of expert observations and a large amount of pre-collected offline (reward-free) data. The goal of *S-off-LfO* is to learn an optimal policy that mimics the provided observations in the environment in which the offline data was collected. Note that here *the environment* that was used to collect the expert observations and *the environment* that was used to collect the offline data are the same environment.

**C-on-LfD.** We have access to a limited number of expert demonstrations and an online interactive training environment. The goal of *C-on-LfD* is to learn an optimal policy that mimics the provided demonstrations in the training environment. Note that here *the environment* that was used to collect the expert demonstrations and *the online training environment* are **not** *the same environment*.

**C-on-LfO.** We have access to a limited number of expert observations (state-only demonstrations) and an online interactive training environment. The goal of *C-on-LfO* is to learn an optimal policy that mimics the provided observations in the training environment. Note that here *the environment* that was used to collect the expert observations and *the online training environment* are **not** *the same environment*.

**C-off-LfD.** We have access to a limited number of expert demonstrations and a large amount of pre-collected offline (reward-free) data. The goal of *C-off-LfD* is to learn an optimal policy that mimics the provided demonstrations in the environment in which the offline data was collected. Note that here *the environment* that was used to collect the expert demonstrations and *the environment* that was used to collect the offline data are **not** *the same environment*.

**C-off-LfO.** We have access to a limited number of expert observations and a large amount of pre-collected offline (reward-free) data. The goal of *C-off-LfO* is to learn an optimal policy that mimics the provided observations in the environment in which the offline data was collected. Note that here *the environment* that was used to collect the expert observations and *the environment* that was used to collect the offline data are **not** *the same environment*.

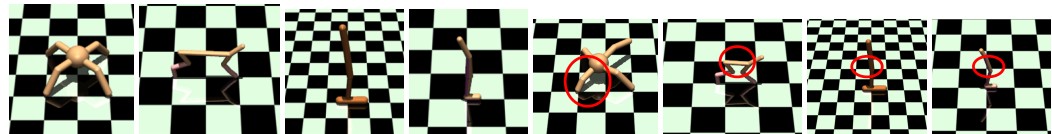

Figure 8: MuJoCo environments and our modified versions. From left to right: Ant-v2, HalfCheetah-v2, Hopper-v2, Walker2d-v2, our modified Ant-v2, our modified HalfCheetah-v2, our modified Hopper-v2, and our modified Walker2d-v2.

## 9.2 Online IL Environments, Offline IL Datasets, and One-shot tasks

Our experiments are conducted in four popular MuJoCo environments (Figure 8): Hopper-v2, HalfCheetah-v2, Walker2d-v2, and Ant-v2. For offline IL tasks, we take the standard (reward-free) D4RL dataset [22] (medium, medium-replay, medium-expert, and expert domains) as the offline dataset. For cross-domain (online/offline) IL tasks, we collect the expert behaviors (demonstrations or observations) on a modified MuJoCo environment. Specifically, we change the height of the agent's torso (as shown in Figure 8). We refer the reader to our code submission, which includes our modified MuJoCo assets. For one-shot IL tasks, we train the policy only in the single-domain IL settings (*S-on-LfD*, *S-on-LfO*, *S-off-LfD*, and *S-off-LfO*). Then we collect only one expert trajectory in the modified MuJoCo environment, and roll out the fine-tuned/inferred policy in the modified environment to test the one-shot performance.

**Collecting expert behaviors.** In our implementation, we use the publicly available rlkit[7] implementation of SAC to learn an expert policy and use the learned policy to collect expert behaviors (demonstrations in LfD or observations in LfO).

## 9.3 CEIL Implementation Details

**Trajectory self-consistency loss.** To learn the embedding function $f_\phi$ and a corresponding contextual policy $\pi_\theta(\mathbf{a}|\mathbf{s}, \mathbf{z})$, we minimize the following trajectory self-consistency loss:

$$\pi_\theta, f_\phi = \min_{\pi_\theta, f_\phi} -\mathbb{E}_{\boldsymbol{\tau}_{1:T} \sim \mathcal{D}(\boldsymbol{\tau}_{1:T})} \mathbb{E}_{(\mathbf{s}, \mathbf{a}) \sim \boldsymbol{\tau}_{1:T}} \left[ \log \pi_\theta(\mathbf{a}|\mathbf{s}, f_\phi(\boldsymbol{\tau}_{1:T})) \right],$$

where $\boldsymbol{\tau}_{1:T}$ denotes a trajectory segment with window size of $T$. In the online setting, we sample trajectory $\boldsymbol{\tau}$ from the experience replay buffer $\mathcal{D}(\boldsymbol{\tau})$; in the offline setting, we sample trajectory $\boldsymbol{\tau}$ directly from the given offline data $\mathcal{D}(\boldsymbol{\tau})$. Meanwhile, if we can access the expert actions (*i.e.*, LfD settings), we also incorporate the expert demonstrations into the empirical expectation (*i.e.*, storing the expert demonstrations into the online/offline experience $\mathcal{D}(\boldsymbol{\tau})$).

In our implementation, we use a 4-layer MLP (with ReLU activation) to encode the trajectory $\boldsymbol{\tau}_{1:T}$ and a 4-layer MLP (with ReLU activation ) to predict the action respectively. To regularize the learning of the encoder function $f_\phi$, we additionally introduce a decoder network (4-layer MLP with ReLU activation) $\pi'_\theta(\mathbf{s}'|\mathbf{s}, f_\phi(\boldsymbol{\tau}_{1:T}))$ to predict the next states: $\min_{\pi'_\theta, f_\phi} -\mathbb{E}_{\boldsymbol{\tau}_{1:T} \sim \mathcal{D}(\boldsymbol{\tau}_{1:T})} \mathbb{E}_{(\mathbf{s}, \mathbf{a}, \mathbf{s}') \sim \boldsymbol{\tau}_{1:T}} \left[ \log \pi'_\theta(\mathbf{s}'|\mathbf{s}, f_\phi(\boldsymbol{\tau}_{1:T})) \right]$. Further, to circumvent issues of "posterior collapse" [67], we encourage learning quantized latent embeddings. In a similar spirit to VQ-VAE [67], we incorporate ideas from vector quantization (VQ) and introduce the following regularization: $\min_{f_\phi} ||\text{sg}[z_e(\boldsymbol{\tau}_{1:T})] - e||^2 + ||z_e(\boldsymbol{\tau}_{1:T}) - \text{sg}[e]||^2$, where $e$ is a dictionary of vector quantization embeddings (we set the size of this embedding dictionary to be 4096), $z_e(\boldsymbol{\tau}_{1:T})$ is defined as the nearest dictionary embedding to $f_\phi(\boldsymbol{\tau}_{1:T})$, and $\text{sg}[\cdot]$ denotes the stop-gradient operator.

**Out-level embedding inference.** In Section 4.2 (main paper), we approximate $\mathcal{J}_{\text{MI}}$ with $\mathcal{J}_{\text{MI}(f_\phi)} \triangleq \mathbb{E}_{p(\mathbf{z}^*)\pi_E(\boldsymbol{\tau}_E)\pi_\theta(\boldsymbol{\tau}_\theta|\mathbf{z}^*)} \left[ -\|\mathbf{z}^* - f_\phi(\boldsymbol{\tau}_E)\|^2 + \|\mathbf{z}^* - f_\phi(\boldsymbol{\tau}_\theta)\|^2 \right]$, where we replace the mutual information with $-\|\mathbf{z}^* - f_\phi(\boldsymbol{\tau})\|^2$ by leveraging the learned embedding function $f_\phi$. Empirically, we find that we can ignore the second loss $\|\mathbf{z}^* - f_\phi(\boldsymbol{\tau}_\theta)\|^2$, and directly conduct outer-level embedding inference with $\max_{\mathbf{z}^*, f_\phi} \mathbb{E}_{p(\mathbf{z}^*)\pi_E(\boldsymbol{\tau}_E)} \left[ -\|\mathbf{z}^* - f_\phi(\boldsymbol{\tau}_E)\|^2 \right]$. Meanwhile, this simplification makes the support constraints ($\mathcal{R}(\mathbf{z}^*)$ in Equation 7 in the main paper) for the offline OOD issues naturally satisfied, since $\max_{\mathbf{z}^*} \mathbb{E}_{p(\mathbf{z}^*)\pi_E(\boldsymbol{\tau}_E)} \left[ -\|\mathbf{z}^* - f_\phi(\boldsymbol{\tau}_E)\|^2 \right]$ and $\min_{\mathbf{z}^*} \mathcal{R}(\mathbf{z}^*)$ are equivalent.

**Cross-domain IL regularization.** To encourage $f_\phi$ to capture the task-relevant embeddings and ignore the domain-specific factors, we set the regularization $\mathcal{R}(f_\phi)$ in Equation 5 to be:

---

[7]https://github.com/rail-berkeley/rlkit.

$\mathcal{R}(f_\phi) = \mathcal{I}(f_\phi(\boldsymbol{\tau}); \mathbf{n})$, where we couple each trajectory $\boldsymbol{\tau}$ in $\{\boldsymbol{\tau}_E\} \cup \{\boldsymbol{\tau}_{E'}\}$ with a label $\mathbf{n} \in \{\mathbf{0}, \mathbf{1}\}$, indicating whether it is noised. In our implementation, we apply MINE [6] to estimate the mutual information and conduct encoder regularization. Specifically, we estimate $\mathcal{I}(\mathbf{z}; \mathbf{n})$ with $\hat{\mathcal{I}}(\mathbf{z}; \mathbf{n}) := \sup_\delta \mathbb{E}_{p(\mathbf{z},\mathbf{n})} [f_\delta(\mathbf{z}, \mathbf{n})] - \log \mathbb{E}_{p(\mathbf{z})p(\mathbf{n})} [\exp(f_\delta(\mathbf{z}, \mathbf{n}))]$ and regularize the encoder $f_\phi$ with $\max_{f_\phi} \hat{\mathcal{I}}(f_\phi(\boldsymbol{\tau}); \mathbf{n})$, where we model $f_\delta$ with a 4-layer MLP (using ReLU activations).

**Hyper-parameters.** In Table 9, we list the hyper-parameters used in the experiments. For the size of the embedding dictionary, we selected it from a range of [512, 1024, 2048, 4096]. We found 4096 to almost uniformly attain good performance across IL tasks, thus selecting it as the default. For the size of the embedding dimension, we tried four values [4, 8, 16, 32] and selected 16 as the default. For the trajectory window size, we tried five values [2, 4, 8, 16, 32] but we did not observe a significant difference in performance across these values. Thus we selected 2 as the default value. For the learning rate scheduler, we tried the default Pytorch scheduler and CosineAnnealingWarmRestarts, and found CosineAnnealingWarmRestarts enables better results (thus we selected it). For other hyperparameters, they are consistent with the default values of most RL implementations, e.g. learning rate 3e-4 and the MLP policy.

Table 9: CEIL hyper-parameters.

| Parameter | Value |
|---|---|
| size of the embedding dictionary | 4096 |
| size of the embedding dimension | 16 |
| trajectory window size | 2 |
| encoder: optimizer | Adam |
| encoder: learning rate | 3e-4 |
| encoder: learning rate scheduler | CosineAnnealingWarmRestarts(T_0 = 1000,T_mult=1, eta_min=1e-5) |
| encoder: number of hidden layers | 4 |
| encoder: number of hidden units per layer | 512 |
| encoder: nonlinearity | ReLU |
| policy: optimizer | Adam |
| policy: learning rate | 3e-4 |
| policy: learning rate scheduler | CosineAnnealingWarmRestarts(T_0 = 1000,T_mult=1, eta_min=1e-5) |
| policy: number of hidden layers | 4 |
| policy: number of hidden units per layer | 512 |
| policy: nonlinearity | ReLU |
| decoder: optimizer | Adam |
| decoder: learning rate | 3e-4 |
| decoder: learning rate scheduler | CosineAnnealingWarmRestarts(T_0 = 1000,T_mult=1, eta_min=1e-5) |
| decoder: number of hidden layers | 4 |
| decoder: number of hidden units per layer | 512 |
| decoder: nonlinearity | ReLU |

Table 10: Baseline methods and their code-bases.

| Baselines | Code-bases |
|---|---|
| GAIL, GAIfO, AIRL | https://github.com/HumanCompatibleAI/imitation |
| SAIL | https://github.com/FangchenLiu/SAIL |
| IQ-Learn, SQIL | https://github.com/Div99/IQ-Learn |
| ValueDICE | https://github.com/google-research/google-research/tree/master/value_dice |
| DemoDICE | https://github.com/KAIST-AILab/imitation-dice |
| SMODICE, ORIL | https://github.com/JasonMa2016/SMODICE |

## 9.4 Baselines Implementation Details

We summarize our code-bases of our baseline implementations in Table 10 and describe each baseline as follows:

**Generative Adversarial Imitation Learning (GAIL).** GAIL [30] is a GAN-based online LfD method that trains a policy (generator) to confuse a discriminator trained to distinguish between generated transitions and expert transitions. While the goal of the discriminator is to maximize the

objective below, the policy is optimized via an RL algorithm to match the expert occupancy measure (minimize the objective below):

$$\mathcal{J}(\pi, D) = \mathbb{E}_\pi\left[\log(D(s, a))\right] + \mathbb{E}_{\pi_E}\left[1 - \log(D(s, a))\right] - \lambda H(\pi).$$

We used the implementation by Gleave et al. [29] on the GitHub page[8], where there are two modifications introduced with respect to the original paper: 1) a higher output of the discriminator represents better, 2) PPO is used to optimize the policy instead of TRPO.

**Generative Adversarial Imitation from Observations (GAIfO).** GAIfO [66] is an online LfO method that applies the principle of GAIL and utilizes a state-only discriminator to judge whether the generated trajectory matches the expert trajectory in terms of states. We provide the objective of GAIfO as follows:

$$\mathcal{J}(\pi, D) = \mathbb{E}_\pi\left[\log(D(s, s'))\right] + \mathbb{E}_{\pi_E}\left[1 - \log(D(s, s'))\right] - \lambda H(\pi).$$

Based on the implementation of GAIL, we implement GAIfO by changing the input of the discriminator to state transitions.

**Adversarial Inverse Reinforcement Learning (AIRL).** AIRL [21] is an online LfD/LfO method using an adversarial learning framework similar to GAIL. It modifies the form of the discriminator to explicitly disentangle the task-relevant information from the transition dynamics. To make the policy more generalized and less sensitive to dynamics, AIRL proposes to learn a parameterized reward function using the output of the discriminator:

$$f_{\theta,\phi}(s, a, s') = g_\theta(s, a) + \lambda h_\phi(s') - h_\phi(s),$$
$$D_{\theta,\phi}(s, a, s') = \frac{\exp(f_{\theta,\phi}(s, a, s'))}{\exp(f_{\theta,\phi}(s, a, s')) + \pi(a|s)}.$$

Similarly to GAIL, we used the code provided by Gleave et al. [29], and the RL algorithm is also PPO.

**State Alignment-based Imitation Learning (SAIL).** SAIL [48] is an online LfO method capable of solving cross-domain tasks. SAIL aims to minimize the divergence between the policy rollout and the expert trajectory from both local and global perspectives: 1) locally, a KL divergence between the policy action and the action predicted by a state planner and an inverse dynamics model, 2) globally, a Wasserstein divergence of state occupancy between the policy and the expert. The policy is optimized using:

$$\mathcal{J}(\pi) = -D_{\mathcal{W}}(\pi(s)\|\pi_E(s)) - \lambda D_{KL}(\pi(\cdot|s_t)\|\pi_E(\cdot|s_t))$$
$$= \mathbb{E}_{\pi(s_t, a_t, s_{t+1})}\left(\sum_{t=1}^{T}\frac{D(s_{t+1}) - \mathbb{E}_{\pi_E(s)}D(s)}{T}\right) - \lambda D_{KL}\left(\pi(\cdot|s_t)\|g_{\text{inv}}(\cdot|s_t, f(s_t))\right),$$

where $D$ is a state-based discriminator trained via $\mathcal{J}(D) = \mathbb{E}_{\pi_E}[D(s)] - \mathbb{E}_\pi[D(s)]$, $f$ is the pretrained VAE-based state planner, and $g_{\text{inv}}$ is the inverse dynamics model trained by supervised regression.

In the online setting, we use the official implementation published by the authors[9], where SAIL is optimized using PPO with the reward definition: $r(s_t, s_{t+1}) = \frac{1}{T}\left[D(s_{t+1}) - \mathbb{E}_{\pi_E(s)}D(s)\right]$. Besides, we further implement SAIL in the offline setting by using TD3+BC [23] to maximize the reward defined above.

In our experiments, we empirically discover that SAIL is computationally expensive. While SAIL is able to learn tasks in the typical IL setting (*S-on-LfD*), our early experimental results find that SAIL(TD3+BC) with heavy hyperparameter tuning failed on the offline setting. This indicates that SAIL is rather sensitive to the dataset composition, which also coincides with the results gathered in Ma et al. [56]. Thus, we do not include SAIL in our comparison results.

**Soft-Q Imitation Learning (SQIL).** SQIL [61] is a simple but effective single-domain LfD IL algorithm that is easy to implement with both online and offline Q-learning algorithms. The main

---

[8]https://github.com/HumanCompatibleAI/imitation
[9]https://github.com/FangchenLiu/SAIL

idea of SQIL is to give sparse rewards (+1) only to those expert transitions and zero rewards (0) to those experiences in the replay buffer. The Q-function of SQIL is updated using the squared soft Bellman Error:

$$\delta^2(\mathcal{D}, r) \triangleq \frac{1}{|\mathcal{D}|} \sum_{(s,a,s') \in \mathcal{D}} \left( Q(s,a) - \left( r + \gamma \log \big( \sum_{a' \in \mathcal{A}} \exp(Q(s',a')) \big) \right) \right)^2.$$

The overall objective of the Q-function is to maximize the following objective:

$$\mathcal{J}(Q) = -\delta^2(\mathcal{D}_E, 1) - \delta^2(\mathcal{D}_\pi, 0).$$

In our experiments, the online imitation policy is optimized using SAC which is also used in the original paper. To make a fair comparison among the offline IL baselines, the offline policy is optimized via TD3+BC.

**Offline Reinforced Imitation Learning (ORIL).** ORIL [78] is an offline single-domain IL method that solves both LfD and LfO tasks. To relax the hard-label assumption (like the sparse rewards made in SQIL), ORIL treats the experiences stored in the replay buffer as unlabelled data that could potentially include both successful and failed trajectories. More specifically, ORIL aims to train a reward function to distinguish between the expert and the suboptimal data without explicitly knowing the negative labels. By incorporating Positive-unlabeled learning (PU-learning), the objective of the reward model can be written as follows (for the LfD setting):

$$\mathcal{J}(R) = \eta \mathbb{E}_{\pi_E(s,a)} \left[ \log(R(s,a)) \right] + \mathbb{E}_{\pi(s,a)} \left[ \log(1 - R(s,a)) \right] - \eta \mathbb{E}_{\pi_E(s,a)} \left[ \log(1 - R(s,a)) \right],$$

where $\eta$ is the relative proportion of the expert data and we set it as 0.5 throughout our experiments. In the original paper, the policy learning algorithm of ORIL is Critic Regularized Regression (CRR), while in this paper, we implemented ORIL using TD3+BC for fair comparisons. Besides, we adapted ORIL to the LfO setting by learning a state-only reward function:

$$\mathcal{J}(R) = \eta \mathbb{E}_{\pi_E(s,s')} \left[ \log(R(s,s')) \right] + \mathbb{E}_{\pi(s,s')} \left[ \log(1 - R(s,s')) \right] - \eta \mathbb{E}_{\pi_E(s,s')} \left[ \log(1 - R(s,s')) \right].$$

**Inverse soft-Q learning (IQ-Learn).** IQ-Learn [28] is an IRL-based method that can solve IL tasks in the online/offline and LfD/LfO settings. It proposes to directly learn a Q-function from demonstrations and avoid the intermediate step of reward learning. Unlike GAIL optimizing a min-max objective defined in the reward-policy space, IQ-Learn solves the expert matching problem directly in the policy-Q space. The Q-function is trained to maximize the objective:

$$\mathbb{E}_{\pi_E(s,a,s')} \left[ Q(s,a) - \gamma V^\pi(s') \right] - \mathbb{E}_{\pi(s,a,s')} \left[ Q(s,a) - \gamma V^\pi(s') - \psi(r), \right.$$

where $V^\pi(s) \triangleq \mathbb{E}_{a \sim \pi(\cdot|s)} \left[ Q(s,a) - \log \pi(a|s) \right]$, $\psi(r)$ is a regularization term calculated over the expert distribution. Then, the policy is learned by SAC.

We use the code provided in the official IQ-learn repository[10] and reproduce the online-LfD results reported in the original paper. For online tasks, we empirically find that penalizing the Q-value on the initial states gives the best and most stabilized performance. The learning objective of the Q-function for the online tasks is:

$$\mathcal{J}(Q) = \mathbb{E}_{\pi_E(s,a,s')} \left[ Q(s,a) - \gamma V^\pi(s') \right] - (1 - \gamma) \mathbb{E}_{\rho_0} \left[ V^\pi(s_0) \right] - \psi(r).$$

In the offline setting, we find that using the above objective easily leads to an overfitting issue, causing collapsed performance. Thus, we follow the instruction provided in the paper and only penalize the expert samples:

$$\mathcal{J}(Q) = \mathbb{E}_{\pi_E(s,a,s')} \left[ Q(s,a) - \gamma V^\pi(s') \right] - \mathbb{E}_{\pi_E(s,a,s')} \left[ V^\pi(s) - \gamma V^\pi(s') \right] - \psi(r)$$
$$= \mathbb{E}_{\pi_E(s,a,s')} \left[ Q(s,a) - V^\pi(s) \right] - \psi(r).$$

**Imitation Learning via Off-Policy Distribution Matching (ValueDICE).** ValueDICE [41] is a DICE-based[11] LfD algorithm which minimizes the divergence of state-action distributions between the policy and the expert. In contrast to the state-conditional distribution of actions $\pi(\cdot|s)$ used in the

---

[10]https://github.com/Div99/IQ-Learn
[11]DICE refers to stationary DIstribution Estimation Correction

above methods, the state-action distribution, $d^\pi(s,a) : \mathcal{S} \times \mathcal{A} \to [0,1]$, can uniquely characterize a one-to-one correspondence,

$$d^\pi(s,a) \triangleq (1-\gamma) \sum_{t=0}^{\infty} \gamma^t \mathbf{Pr}(s_t = s, a_t = a \,| s_0 \sim \rho_0, a_t \sim \pi(s_t), s_{t+1} \sim P(s_t, a_t)).$$

Thus, the plain expert matching objective can be reformulated and expressed in the Donsker-Varadhan representation:

$$\begin{aligned}
\mathcal{J}(\pi) &= -D_{KL}(d^\pi(s,a) \| d^{\pi_E}(s,a)) \\
&= \min_{x:\mathcal{S} \times \mathcal{A} \to \mathbb{R}} \log \mathbb{E}_{(s,a) \sim d^{\pi_E}} \left[ \exp(x(s,a)) \right] - \mathbb{E}_{(s,a) \sim d^\pi} \left[ x(s,a) \right].
\end{aligned}$$

The objective above can be expanded further by defining $x(s,a) = v(s,a) - \mathcal{B}^\pi v(s,a)$ and using a zero-reward Bellman operator $\mathcal{B}^\pi$ to derive the following (adversarial) objective:

$$\mathcal{J}_{DICE}(\pi, v) = \log \mathbb{E}_{(s,a) \sim d^{\pi_E}} \left[ \exp\left(v(s,a) - \mathcal{B}^\pi v(s,a)\right) \right] - (1-\gamma)\mathbb{E}_{s_0 \sim \rho_0, a_0 \sim \pi(\cdot|s_0)} \left[ v(s_0, a_0) \right].$$

We use the official Tensorflow implementation[12] in our experiments. In the online setting, the rollouts collected are used as an additional replay regularization. The overall objective in the online setting is:

$$\begin{aligned}
&\mathcal{J}_{DICE}^{mix}(\pi, v) \\
&= -D_{KL}\big((1-\alpha)d^\pi(s,a) + \alpha d^{RB}(s,a) \| (1-\alpha)d^{\pi_E}(s,a) + \alpha d^{RB}(s,a)\big) \\
&= \log \mathbb{E}_{(s,a) \sim d^{mix}} \left[ \exp\left(v(s,a) - \mathcal{B}^\pi v(s,a)\right) \right] - (1-\alpha)(1-\gamma) \, \mathbb{E}_{s_0 \sim \rho_0, \, a_0 \sim \pi(\cdot|s_0)} \left[ v(s_0, a_0) \right] \\
&\qquad\qquad\qquad\qquad\qquad\qquad\qquad\qquad - \alpha \, \mathbb{E}_{(s,a) \sim d^{RB}} \left[ v(s,a) - \mathcal{B}^\pi v(s,a) \right],
\end{aligned}$$

where $d^{mix} \triangleq (1-\alpha)d^{\pi_E} + \alpha d^{RB}$ and $\alpha$ is a non-negative regularization coefficient (we set $\alpha$ as 0.1 following the specification of the paper).

In the offline setting, ValueDICE only differs in the source of sampling data. We change the online replay buffer to the offline pre-collected dataset.

**Offline Imitation Learning with Supplementary Imperfect Demonstrations (DemoDICE).**
DemoDICE [39] is a DICE-based offline LfD method that assumes to have access to an offline dataset collected by a behavior policy $\pi_\beta$. Using this supplementary dataset, the expert matching objective of DemoDICE is instantiated over ValueDICE:

$$-D_{KL}(d^\pi(s,a) \| d^{\pi_E}(s,a)) - \alpha D_{KL}(d^\pi(s,a) \| d^{\pi_\beta}(s,a)),$$

where $\alpha$ is a positive weight for the constraint.

The above optimization objective can be transformed into three tractable components: 1) a reward function $r(s,a)$ derived by pre-training a binary discriminator $D : \mathcal{S} \times \mathcal{A} \to [0,1]$:

$$\begin{aligned}
r(s,a) &= -\log(\frac{1}{D^*(s,a)} - 1), \\
D^*(s,a) &= \arg\max_D = \mathbb{E}_{d^{\pi_E}} \left[ \log D(s,a) \right] + \mathbb{E}_{d^{\pi_\beta}} \left[ \log(1 - D(s,a)) \right],
\end{aligned}$$

2) a value function optimization objective:

$$\mathcal{J}(v) = -(1-\gamma)\mathbb{E}_{s \sim \rho_0} \left[ v(s) \right] - (1+\alpha) \log \mathbb{E}_{(s,a) \sim d^{\pi_\beta}} \left[ \exp(\frac{r(s,a) + \mathbb{E}_{s' \sim P(s,a)}(v(s')) - v(s)}{1+\alpha}) \right],$$

and 3) a policy optimization step:

$$\begin{aligned}
\mathcal{J}(\pi) &= \mathbb{E}_{(s,a) \sim d^{\pi_\beta}} \left[ v^*(s,a) \log \pi(a|s) \right], \\
v^*(s,a) &= \arg\max_v \mathcal{J}(v).
\end{aligned}$$

We report the offline results using the official Tensorflow implementation[13].

---

[12]https://github.com/google-research/google-research/tree/master/value_dice
[13]https://github.com/KAIST-AILab/imitation-dice

**State Matching Offline DIstribution Correction Estimation (SMODICE).** SMODICE [56] proposes to solve offline IL tasks in LfO and cross-domain settings and it optimizes the following state occupancy objective:

$$-D_{KL}(d^{\pi}(s)\|d^{\pi_E}(s)).$$

To incorporate the offline dataset, SMODICE derives an f-divergence regularized state-occupancy objective:

$$\mathbb{E}_{s\sim d^{\pi}(s)}\left[\log(\frac{d^{\pi_\beta}(s)}{d^{\pi_E}(s)})\right] + -D_f(d^{\pi}(s,a)\|d^{\pi_\beta}(s,a)).$$

Intuitively, the first term can be interpreted as matching the offline states towards the expert states, while the second regularization term constrains the policy close to the offline distribution of state-action occupancy. Similarly, we can divide the objective into three steps: 1) deriving a state-based reward by learning a state-based discriminator:

$$r(s,a) = -\log(\frac{1}{D^*(s)} - 1),$$
$$D^*(s,a) = \arg\max_{D} = \mathbb{E}_{d^{\pi_E}}\left[\log D(s)\right] + \mathbb{E}_{d^{\pi_\beta}}\left[\log(1 - D(s))\right],$$

2) learning a value function using the learned reward:

$$\mathcal{J}(v) = -(1-\gamma)\mathbb{E}_{s\sim\rho_0}\left[v(s)\right] - \log\mathbb{E}_{(s,a)\sim d^{\pi_\beta}}\left[f_*(r(s,a) + \mathbb{E}_{s'\sim P(s,a)}(v(s')) - v(s))\right],$$

and 3) training the policy via weighted regression:

$$\mathcal{J}(\pi) = \mathbb{E}_{(s,a)\sim d^{\pi_\beta}}\left[f'_*(r(s,a) + \mathbb{E}_{s'\sim P(s,a)}(v^*(s')) - v^*(s))\log\pi(a|s)\right],$$
$$v^*(s,a) = \arg\max_{v}\mathcal{J}(v),$$

where $f_*$ is the Fenchel conjugate of f-divergence (please refer to Ma et al. [56] for more details).

We conduct experiments using the official Pytorch implementation [14], where the f-divergence used is $\mathcal{X}^2$-divergence. On the LfD tasks, we change the input of the discriminator to state-action pairs.

---

[14]https://github.com/JasonMa2016/SMODICE

