# OpenReview forum: "CEIL: Generalized Contextual Imitation Learning"
_NeurIPS.cc/2023/Conference — NeurIPS 2023 poster_

### Official Review · Reviewer_STjS · 2023-06-22

**Soundness:** 3 good
**Presentation:** 4 excellent
**Contribution:** 3 good
**Rating:** 6
**Confidence:** 4

**Summary:**

This paper aims to develop a simple and scalable IL method, which is applicable to a wide range of IL settings (e.g., offline/online, LfD/LfO, etc.). To achieve this,  the imitation policy are decoupled into a contextual policy and a latent variable. Experiments on a wide range of tasks illustrate effectiveness of the proposed methods.

**Strengths:**

1. This paper is well written and easy to follow. I enjoy reading it.
2. Authors conduct extensive experiments. It is a solid work.
3. I notice that authors present a detailed implementation details on baselines (like GAIL, GAIfO, et.) in the appendix, which will serve as a good reference for imitation learning researchers.
4. The proposed framework is simple and compatible with many different IL settings.

**Weaknesses:**

1. Results of online HalfCheetah seems to be lost.
2. Adding comparisons with recently popular (decision-)transformer-style methods in their compatible settings (e.g., [1] for offline IL, [2] for offline multi-task IL, [3] for one-shot IL and LfO) will make this work more promising.
3. The proposed method, CEIL, is not a plug-in method for every IL setting. We still need to make considerate modifications and designations when meeting new tasks. And a lot of works upon the generative adversarial imitation learning (GAIL) framework has been proposed to deal with different IL settings and they work well (e.g., [4] for cross-domain IL, [5] for one-shot IL). A further discussion on the superiority of this work over existing GAIL-style works will be appreciated.

[1] Carroll, Micah, et al. Unimask: Unified inference in sequential decision problems. arXiv preprint arXiv:2211.10869 (2022).

[2] Furuta, Hiroki, Yutaka Matsuo, and Shixiang Shane Gu. Generalized decision transformer for offline hindsight information matching. arXiv preprint arXiv:2111.10364. 2021.

[3] Xu, Mengdi, et al. Hyper-decision transformer for efficient online policy adaptation. arXiv preprint arXiv:2304.08487 (2023).

[4] Franzmeyer, Tim, Philip Torr, and João F. Henriques. Learn what matters: cross-domain imitation learning with task-relevant embeddings. Advances in Neural Information Processing Systems 35 (2022): 26283-26294.

[5] Yu, Lantao, et al. Meta-inverse reinforcement learning with probabilistic context variables. Advances in neural information processing systems 32 (2019).

**Questions:**

1. I read the derivation of Equation (6) in the appendix and get confused on the constant $C$. If I am not wrong,
$$
C = \mathbb{E}_{p(z^*)\pi_E(\tau)}\left[\log \frac{\pi_\theta(\tau)}{\tau_E(\tau)}\right] = \mathbb{E}_{p(z^*)\pi_E(\tau)}\left[\log \frac{\mathbb{E}_{p(z^*)}[\pi_\theta(\tau|z^*)]}{\tau_E(\tau)}\right],
$$
how can it be a constant to $z^*$? By the way, what do you mean by $p(z^*)$? $z^*$ is a single variable not a distribution that you plan to optimize, right?
2. Considering the one-shot IL setting and your claim in line 246-248, I wonder how you guarantee that $f_\phi$ will generate a proper latent embedding? Will it be trained in a meta-style when meeting one-shot IL tasks?

I am willing to raise my score if you can answer my questions and solve my concerns. Thanks in advance.

**Limitations:**

This works does not seem to have any negative societal impact. Authors discussed its limitations and planned to leave them for future work.

---

> ### Author Rebuttal · Authors · 2023-08-08
>
> We thank the reviewer for the detailed and thoughtful feedback. We will address your concerns one by one.
>
> **Q1: results of online HalfCheetah.**
>
> **A1:** Thank you for the suggestion. We have provided online HalfCheetah results, Figure 3, and more comparison results in Atari and Adroit domain, Figure 1, in the PDF file in the general response and we will add it in our revision.
>
> **Q2: comparisons with recently popular (decision-)transformer-style methods.**
>
> **A2:** Thank you for the suggestion. We have conducted new comparison against Unimask and GDT (generalized decision transformer). Please refer to the PDF file (Table 2) in the "global" response. We can see that CEIL consistently performs better than or comparable to Unimask and GDT.
>
>
> **Q3: the superiority of this work over existing GAIL-style works.**
>
> **A3:** Thank you for raising this concern. By comparing CEIL with the plain expert matching objective $\min_\theta D(\pi_\theta(\tau), \pi_E(\tau))$ in existing GAIL-style works, we highlight two merits: 1) CEIL’s expert matching loss does not account for updating $\pi_\theta$ and is only incentivized to update the low-dimensional latent variable $z*$, which enjoys efficient parameter learning similar to the prompt tuning in large language models, and 2) we learn $\pi_\theta$ by simply performing supervised regression, which is more stable compared to vanilla inverse-RL/adversarial-IL (GAIL-style) methods.
>
> **Q4: the constant C in the derivation in the appendix.**
>
> **A4:** Thank you for the careful review. It is a typo. In Equation 6 and its derivation in the appendix, 'D_{KL}' should be 'D', where we defined 'D' as the sum of ‘D_{KL}’ and inverse 'D_{KL}' (see Footnote 3 in Page 5, main paper). Therefore, there is no constant C in the corresponding derivation. We will correct it. Thank you.
>
> **Q5: $z\*$ is a single variable in the implementation, right?**
>
> **A5:** Yes, it is.
>
> **Q6: Will it be trained in a meta-style when meeting one-shot IL tasks?**
>
> **A6:** In response to your concern, we are actually training in a meta-style. Intuitively, Equation 3 treats each trajectory individually as a separate task (a separate z for each trajectory/task), in the same spirit of meta-learning. As the reviewer points out, our approach can also be adapted to multi-task IL and one-shot IL tasks, benefiting from the separating optimization of $\pi_\theta$ and $z*$.
> As for the generalization, how well the model generalizes depends on the (multi-task) data distribution $D(\tau)$.
>
> Thanks again for all of your constructive suggestions, which have helped us improve the quality and clarity of our paper.

---

> > ### Comment · Reviewer_STjS · 2023-08-14
> > **Response to Rebuttal**
> >
> > Thanks to your response, which generally solves my concerns.
> >
> > I notice that recently there emerge some works on optimizing a contextual variable in reinforcement learning, thus extending this idea to imitation learning is natural. I appreciate the contribution of this work, but as I pointed in Weakness 3, there may be still a lot of work to do to figure out how to get $z^*$ when meeting different problem settings, leaving space for further investigation.
> >
> > I raise my score from 5 to 6.

---

### Official Review · Reviewer_2c9A · 2023-07-02

**Soundness:** 3 good
**Presentation:** 4 excellent
**Contribution:** 3 good
**Rating:** 6
**Confidence:** 4

**Summary:**

The objective of this work is to create an imitation learning (IL) method that functions effectively across a variety of common IL settings, which include online, offline, Learning from Demonstrations (LFD), Learning from Observation (LFO), and cross-domain. To achieve this, a dual-level expert matching goal is proposed. This involves not only trajectory matching with expert demonstration but also matching the latent variable, z, with the expert in latent space.

Contrasting with traditional trajectory matching methods, matching expert trajectories in latent space appears to more accurately replicate expert behavior. Experimental results reinforce the efficacy of this methodology, demonstrating good performance across four different MuJoCo tasks examined in this study.

**Strengths:**

- The proposed method can adjust to five different Interactive Learning (IL) settings with mere minor adjustments in the algorithm, lending the model practical applications in real-world scenarios.

- The document provides an extensive review of existing methodologies across these five IL settings, thereby allowing for a broad understanding of the field.

- Additionally, by comparing the suggested methods against multiple baselines, good results have been demonstrated.
The presentation of the paper communicates ideas clearly.


**Weaknesses:**

- The claim that all Online, Offline, LFD, LFO, and cro-domain issues are addressed appears to be overly ambitious. Given that only two MuJoCo tasks were evaluated in an online setting and four in an offline setting, the experimental evidence provided is inadequate to substantiate this claim.


- Furthermore, the conducted ablation study does not provide sufficient insights into the effectiveness of the J_MI term, which is designed to align with the expert demonstration in the latent space. To magnify our understanding of J_MI's contribution, an additional ablation baseline CEIL without the J_MI term could be quite useful. If, subsequently, we find that the removal of J_MI leads to a performance reduction of 10%, 30%, or even more, we can then quantify the effectiveness of matching expert demonstrations in the latent space with the specific technique used in this work. Providing such a comparison can significantly enhance the reader's understanding of the extent of performance improvements achieved through the application of J_MI.

**Questions:**

''Offline results generally outperform online results, especially in the LfO setting. ''

Q1: Can you provide more details on this particular issue? Under similar conditions with the same volume of expert demonstration data, the online setting is generally capable of accessing more information. Therefore, it's not quite clear why the proposed method would underperform in an online setting.

Q2: Similar to bi-level learning methods such as GAIL and IRLs, they first learn a discriminator or reward function, followed by policy training based on the learned discriminator or reward function. Frequently, this type of structure tends to struggle with instability and prolonged training durations. Does the proposed bi-level training process encounter similar issues and why?

**Limitations:**

See weakness and questions.

---

> ### Author Rebuttal · Authors · 2023-08-08
>
> We thank the reviewer for the detailed and thoughtful feedback. We will address your concerns one by one.
>
> **Q1: the experimental evidence.**
>
> **A1:** We have carried out new experiments in Atari and Adroit domains (see results, Figure 1, in the PDF file in the "global" response). We can see that CEIL consistently achieves better or comparable performance compared to baseline methods in both online and offline IL tasks.
>
> **Q2: an additional ablation baseline CEIL without the J_MI term.**
>
> **A2:** Thank you for such a valuable suggestion for improving this paper. We have carried out new ablation experiments on the J_MI term in both online IL and offline IL settings (see results, Figure 2 and Table 1, in the PDF file in the general response). We can see that ablating J_MI does lead to degraded performance, further verifying the effectiveness of our expert matching objective in the latent space.
>
> **Q3: offline results generally outperform online results, especially in the LfO setting.**
>
> **A3:** Yes. In Appendix 2.6 (supplementary material), we also find this limitation in the online LfO setting. We believe this is due to the lack of explicit exploration bonus, which causes the agent to stay in the collapsed state region and therefore deteriorates performance. To address this limitation, here we explicitly impose a lower bound regularization on the policy entropy to encourage exploration, borrowing the idea from SAC and online Decision Transformer [1]. We implement such a regularization with the Lagrangian relaxation method. We refer the reviewer to our new results (Figure 3) in the PDF file in the "global" response, where we find such a simple exploration strategy can effectively bridge the gap between online IL and offline IL, improving the online LfO performance and outperforming online IL baselines.
>
> [1] Zheng, Qinqing, Amy Zhang, and Aditya Grover. "Online decision transformer." international conference on machine learning. PMLR, 2022.
>
> **Q4: Does the proposed bi-level training process encounter similar issues (in GAIL and IRLs) and why?**
>
> **A4:** No, it does not. Many existing (GAIL or IRLs) methods are difficult to train in practice due to an adversarial optimization process over reward and policy approximators (biased or high variance gradient estimators). Another challenge in the GAN-style objective is balancing the performance of the generator (policy) and discriminator. A discriminator that achieves very high accuracy can produce relatively uninformative gradients, but a weak discriminator can also hamper the policy’s ability to learn. However, CEIL learns (a contextual) policy by simply performing supervised regression, which is more stable compared to vanilla GAIL and IRLs. Further, to recover the expert behaviors, CEIL's expert matching objective is optimized over the representation space, which does not depend on the accuracy of the policy. This benefit is even more significant in cross-domain IL settings, since we only need a good representation to characterize trajectories.
>
> Thanks again for all of your constructive suggestions, which have helped us improve the quality and clarity of our paper.

---

> > ### Comment · Reviewer_2c9A · 2023-08-17
> > **Reply to Author**
> >
> > Thank you for the rebuttal! My concerns have been addressed.

---

### Official Review · Reviewer_7vt9 · 2023-07-05

**Soundness:** 3 good
**Presentation:** 3 good
**Contribution:** 3 good
**Rating:** 6
**Confidence:** 3

**Summary:**

This work proposed ContExtual Imitation Learning (CEIL), a general method that can be applied to multiple settings, including learning from observations (LfO), offline IL, cross-domain IL, and one-shot IL. CEIL incorporates the hindsight information-matching principle within a bi-level expert matching objective, which decouples the learning policy into a contextual policy and an optimal embedding. Empirical analysis demonstrates that CEIL is more sample efficient in online IL and performs well in offline IL settings.

**Strengths:**

1. CEIL is closely related to hindsight information-matching methods. CEIL introduces an additional context variable $z$ to learn a contextual policy $\pi_\theta(a|s, z)$ and an optimal contextual variable $z^*$. The idea is to use the learned $z^*$-conditioned policy $\pi_\theta(a|s, z^*)$ to recover the expert data. CEIL is learned through a bi-level expert matching objective: explicitly learn a hindsight embedding function in the inner-level optimization, and perform expert matching via inferring an optimal embedding in the outer-level optimization. Such a decoupling procedure enables CEIL to generalize to diverse IL settings.
2. Unlike the prior hindsight information-matching methods, CEIL does not require explicit handling components such as explicit rewards in online RL and handcrafted target return in offline RL.
3. CEIL is a scalable method that can be applied to LfD, LfO, offline IL, cross-domain IL, and one-shot IL. CEIL was evaluated in diverse settings in the experiments. The results on four MuJoCo environments show that CEIL achieves better sample efficiency than other baselines in the online IL tasks. Extensive ablations demonstrated that CEIL is not sensitive to the number of demonstrations and window size of trajectory.


**Weaknesses:**

1. It seems that the pre-defined return $f_R(\tau)$ in Equation (2) is closely related to the hindsight embedding function $f_\phi(\tau)$ in Equation (3). However, their relationship has not been discussed.
2. In the cross-domain online LfO experiments (Hopper, Figure 2(d)), CEIL performs worse than AIRL (state only). There is no explanation for this result.


**Questions:**

1. What is the relationship between the return $f_R(\tau)$ in Equation (2) and the hindsight embedding function $f_\phi(\tau)$ in Equation (3)?
2. In Equation (7), what is the motivation for applying stop gradient operation to $f_{\bar{\phi}}$? Is there any guarantee that this operation will satisfy the support in Equation (5)?
3. Why does CEIL perform worse than AIRL in the cross-domain online LfO task (Hopper, Figure 2(d))?


**Limitations:**

1. CEIL lacks explicit exploration bounds; thus, the offline results generally outperform the online results, especially in the LfO setting.
2. The trajectory self-consistency cannot be applied to cross-embodiment agents once the two embodiments/domains have different state spaces or action spaces.


---------------------
After rebuttal
---------------------
Thanks to the authors for their rebuttal. I'd like to increase my score to 6.

---

> ### Author Rebuttal · Authors · 2023-08-08
>
>
> We thank the reviewer for the detailed and thoughtful feedback. We will address your concerns one by one.
>
> **Q1: the relationship between the pre-defined return $f_R(\tau)$ in Equation 2 and $f_\phi(\tau)$ in Equation 3.**
>
> **A1:** Compared to the pre-defined return $f_R(\tau)$, $f_\phi(\tau)$ can be seen as a more general formulation representing the hindsight information of a trajectory. $f_\phi(\tau)$ could be any function of a trajectory that captures some statistical properties in state-space or trajectory-space, such as sufficient statistics of a distribution, such as mean, variance or higher-order moments [1]. Empirically, we also find that trajectories with high $f_R(\tau)$ are closer to the encodings of expert trajectories in the latent space (as implied by Figure 1 in the main paper). Thank you for the valuable suggestion, we will add new discussions in our paper revision.
>
> [1] Furuta H, Matsuo Y, Gu S S. Generalized decision transformer for offline hindsight information matching[J]. arXiv preprint arXiv:2111.10364, 2021.
>
> **Q2: a stop gradient operation to $f_{\bar{\phi}}$ in Equation 7. The guarantee for the support constraint.**
>
> **A2:** In Equation 7, we are optimizing $z*$, thus we add the stop gradient operator to $f_{\bar{\phi}}$. The guarantee for the support constraint comes from BCQ [2]. Intuitively, Equation 7 minimizes the distance of selected $z*$ to the embeddings of the batch data ($\tau_E$ and $\tau_D$), forcing $z*$ towards behaving close to a subset of the offline and expert behaviors. The main difference is that BCQ implements offline support constraint over the action space while we restrict it over the latent space.
>
> [2] Fujimoto S, Meger D, Precup D. Off-policy deep reinforcement learning without exploration[C]//International conference on machine learning. PMLR, 2019: 2052-2062.
>
> **Q3: Why does CEIL perform worse than AIRL in the cross-domain online LfO task (Hopper, Figure 2(d))?**
>
> **A3:** As implied by the reviewer, we believe that this is due to the lack of explicit exploration bonus, which causes the agent to stay in the collapsed state region and therefore deteriorates performance. To address this limitation, we have explicitly imposed a lower bound regularization on the policy entropy to encourage exploration, borrowing the idea from SAC and online Decision Transformer [3]. We implement such regularization with the Lagrangian relaxation method. We refer the reviewer to our new results (Figure 3) in the PDF file in the general response, where we find such a simple exploration strategy can effectively bridge the gap between online IL and offline IL, improving the online LfO performance and outperforming online LfO baselines.
>
> [3] Zheng, Qinqing, Amy Zhang, and Aditya Grover. "Online decision transformer." international conference on machine learning. PMLR, 2022.
>
> **Q4: the two embodiments/domains have different state spaces or action spaces.**
>
> **A4:** We appreciate your concern about the state/action space. If cross-domain agents have different action/state spaces, a typical approach is to serialize state/action from different modalities into a flat sequence of tokens [4]. We remark that CEIL is also compatible with such a tokenization approach, thus suitable for IL tasks with different actions/state spaces, yet it is out of the scope of the current work.
>
> [4] Reed S, Zolna K, Parisotto E, et al. A generalist agent[J]. arXiv preprint arXiv:2205.06175, 2022.
>
> Thanks again for all of your constructive suggestions, which have helped us improve the quality and clarity of our paper.

---

> > ### Comment · Reviewer_7vt9 · 2023-08-17
> > **Reply to author**
> >
> > Thank you for the rebuttal. I would like to increase my score to 6.

---

### Official Review · Reviewer_PiwE · 2023-07-06

**Soundness:** 3 good
**Presentation:** 3 good
**Contribution:** 4 excellent
**Rating:** 6
**Confidence:** 4

**Summary:**

This paper presents a method that aims to address Imitation Learning (IL) tasks by simultaneously updating the embedding function of a contextual variable, an optimal contextual variable, and a policy conditioned on that variable. The proposed method learns the conditional policy by minimizing the trajectory self-consistency loss based on the concept of hindsight information matching. The optimal contextual variable is updated by minimizing the discrepancy between the learned conditional policy and the expert policy. The embedding function is optimized based on both the self-consistency loss and the discrepancy loss. The experimental results demonstrated improved performance in the following tasks: (1) learning from observations (LfO), (2) online/offline IL, (3) cross-domain IL, and (4) one-shot generalization IL.

**Strengths:**

1. The proposed method is novel, and enables solving a variety of IL tasks with minimal adjustments.
2. The experiments were conducted on 8 different IL settings and outperformed previous baselines in most environments.
3. The empirical analysis in Section 5.2 offers interesting insights into the practical application of the proposed method.

**Weaknesses:**

1. A number of hyperparameters are introduced in this work, such as the embedding dimension, the trajectory window size, the architecture of the encoder/decoder networks. However, the authors do not provide any guidance or recommendations regarding the selection of these hyperparameters.
2. There appears to be a disparity between the theoretical objective (Eq.(4), Eq.(5)) and the practical objective (Eq.(8), and Line 6 in Algorithm 1) when it comes to optimizing the hindsight embedding function $f_{\phi}$. Theoretically, the update of $f_{\phi}$ should solely be based on the trajectory self-consistency loss, as mentioned in Eq.(4). In practice, however, $f_{\phi}$ is also updated according to Eq.(8).
3. The notation for the regularization losses is ambiguous. Specifically, there are two regularization losses used in this work, one for regularizing the embedding function $f_{\phi}$ (Eq.(9)), and another for regularizing the optimal contextual variable $\mathbf{z}^*$ (Eq.(7)). Unfortunately, both of these losses are represented by the symbol $\mathcal{R}$, and their definitions are scattered throughout the manuscript, resulting in unnecessary confusion.
4. The confidence intervals are not reported in Table 2-4.
5. The expert trajectories generated by SAC used in the experiments have not been provided, which could potentially pose challenges for replicating the results in future studies.

**Questions:**

1. As mentioned in Weakness (1), how are the hyperparameters chosen in the experiments? Could the authors offer guidance on the process of selecting the hyperparameters (those in Appendix Table 4)?
2. As stated in Weakness (2), could the authors provide ablation studies and further clarification on the optimization of $f_{\phi}$ as described in Eq.(8)?
3. As mentioned in Weakness (3), could the authors clarify that there exist two regularization losses in this work? If possible, it would be best to include both $\mathcal{R}(f_{\phi})$ and $\mathcal{R}(\mathbf{z}^*)$ in Eq.(5). Alternatively, the authors could at least mention (in Line 164-165) that $\mathbf{z}^* \in  f_{\phi}\circ\mathrm{supp}(\mathcal{D})$ is achieved through minimizing $\mathcal{R}(\mathbf{z}^*)$. It is acceptable to use the symbol $\mathcal{R}$ to represent both losses, as long as it is straightforward for readers to distinguish between the two.

**Limitations:**

Minor suggestions:
- Section 3.1, Line 93: The transition dynamics function should be $\mathcal{T}$, not $\mathcal{P}$.
- Section 3.1, Line 98: The transition dynamics function should be $\mathcal{T}$, not $T$.
- Section 3.2, Line 124: Missing citation for hindsight experience replay (HER) [[1]].
- Algorithm 1, Line 4: The two $\mathcal{D}$ here can be combined into a single $\mathcal{D}$ for simplicity.

[1]: https://arxiv.org/abs/1707.01495

---

> ### Author Rebuttal · Authors · 2023-08-08
>
> We thank the reviewer for the detailed and thoughtful feedback. We will address your concerns one by one.
>
> **Q1: guidance on the process of selecting the hyperparameters in the Appendix.**
>
> **A1:** For the size of the embedding dictionary, we selected it from a range of [512, 1024, 2048, 4096]. We found 4096 to almost uniformly attain good performance across IL tasks, thus selecting it as the default. For the size of the embedding dimension, we tried four values [4, 8, 16, 32] and selected 16 as the default. For the trajectory window size, we tried five values [2, 4, 8, 16, 32] but we did not observe a significant difference in performance across these values. Thus we selected 2 as the default value. For the learning rate scheduler, we tried the default Pytorch scheduler and CosineAnnealingWarmRestarts, and found CosineAnnealingWarmRestarts enables better results (thus we selected it). For other hyperparameters, they are consistent with the default values of most RL implementations, e.g. learning rate 3e-4 and the MLP policy.
>
> **Q2: ablation studies on the optimization of $f_\phi$.**
>
> **A2:** Thank you for such a valuable suggestion. We have carried out new ablation experiments on the loss of $f_\phi$ in both online IL and offline IL settings (see results, Figure 2 and Table 1, in the PDF file in the "global" response). We can see that ablating the $f_\phi$ loss (optimizing $f_\phi$ with Equation 5) does degrade the performance in both online and offline IL tasks, demonstrating the effectiveness of optimizing $f_\phi$ with Equation 8. Intuitively, Equation 8 encourages the embedding function $f_\phi$ to be task-relevant, and thus we use the expert matching loss to update $f_\phi$.
>
> **Q3: the notation for the regularization losses.**
>
> **A3:** We appreciate your suggestion. Actually, the support constraint is achieved through minimizing $R(z*)$. There is only one regularization (for cross-domain IL) in this work. We will clarify it in our paper revision.
>
> **Q4: confidence intervals, expert data, and other suggestions.**
>
> **A4:** Thank you for the valuable suggestions. We will incorporate/elaborate on them in our revision.
>
> Thanks again for all of your constructive suggestions, which have helped us improve the quality and clarity of our paper.

---

> > ### Comment · Reviewer_PiwE · 2023-08-15
> >
> > I've read the authors' rebuttal and will maintain my score of 6.

---

### Official Review · Reviewer_vVku · 2023-07-08

**Soundness:** 3 good
**Presentation:** 3 good
**Contribution:** 3 good
**Rating:** 7
**Confidence:** 3

**Summary:**

One recent idea in reinforcement learning is to learn a sequential model that can predict state-action transitions and rewards, and then obtain a strong policy by inferring actions conditioned on high reward. This has the major benefit that even low-reward trajectories provide useful information for learning.

The challenge in adapting this idea to imitation learning is that in imitation learning, we do not have access to rewards, but rather a dataset of expert demonstrations, plus (potentially) datasets of other suboptimal behavior (e.g. rollouts from a policy in the online setting, static dataset from suboptimal policies in the offline setting).

The proposed solution is to simply infer / learn a contextual variable $z$ that performs a similar role as reward: that is, it determines the “type” of policy / actions that make up the trajectory. To avoid $z$ from encoding too much information, we make it very limited and regularize it. We can then train a $z$-conditional policy that can produce actions that mimic both the suboptimal behavior as well as the expert demonstrations (using a self-consistency loss). We also learn an optimal setting $z^*$ that selects the high-performing policy, by ensuring that the $z^*$-conditional policy produces trajectories with high similarity to the expert trajectories. There is a significant amount of math to flesh this out, which I won’t get into.

The authors test their method on four MuJoCo environments, using expert demonstrations from a SAC policy, in both online / offline settings, in both full demos / observation-only settings, and with / without a distributional shift (modifying the torso length) in the test environment, and show that CEIL tends to match or improve upon the results from a variety of baselines.

**Strengths:**

1. The area of imitation learning is important and relevant, and the approach suggested covers a wide variety of use cases.
2. The idea proposed is conceptually simple: train a network that models a variety of policies, and then select an appropriate high-performing policy through the use of context variable that controls the policy. Similar ideas have seen significant success in reinforcement learning.
3. The empirical evaluations are quite extensive with an impressive number of baselines, and show strong performance of the author’s method in a wide variety of settings.
4. I particularly appreciated Table 1, which provides an excellent overview of related work.
5. While I found the paper hard to read in an absolute sense, I think that is mostly because the ideas are quite technical: relative to other imitation learning papers focused on expert matching, I found this paper easier to read and understand.

Overall, I recommend accepting the paper. However, I should note that I am not very familiar with the imitation learning literature, and so cannot provide an evaluation of the following aspects:
1. How novel / original these ideas are
2. Whether the baselines chosen are appropriate (e.g. perhaps the paper has not compared to current SOTA)
3. Whether the performance of baselines is in line with expected numbers (e.g. perhaps the authors did not tune hyperparameters of baselines well)

**Weaknesses:**

1. The empirical evaluations are entirely based on MuJoCo. It is unclear whether the strong performance will generalize to very different settings (e.g. Atari).
2. Though the paper aims to provide a general and broadly applicable imitation learning algorithm, it doesn’t tackle the most typical imitation learning setting: where we have access only to a dataset of expert demonstrations. (In particular, even in the offline setting, CEIL assumes access to a dataset of suboptimal behavior.) In principle we could run the algorithm in such a setting, though my guess is that it will not perform as well as baseline methods like behavior cloning, since the major benefit of CEIL is in its ability to leverage suboptimal data. (However, the expert-demos-only setting tends to be very vulnerable to spurious correlations / overfitting, and so is not as significant as the other settings.)

**Questions:**

How would you expect CEIL to work in settings other than MuJoCo?

Have you run CEIL with access only to expert demonstrations? How does it perform in that setting?

**Limitations:**

The paper should mention that it does not tackle the setting in which we only have expert demonstrations (even in the offline setting, the paper assumes access to a static dataset of suboptimal behavior, specifically D4RL in its experiments).

---

> ### Author Rebuttal · Authors · 2023-08-08
>
> We thank the reviewer for the detailed and thoughtful feedback. We will address your concerns one by one.
>
> **Q1: whether the strong performance will generalize to very different settings (e.g. Atari).**
>
> **A1:** As suggested by the reviewer, we have carried out new experiments on Atari and Adroit domains. We refer the reviewer to our new results (Figure 1) in the PDF file in the general response. We can see that CEIL consistently achieves better or comparable performance compared to baseline methods in both online (Atari) and offline (Adroit) IL tasks.
>
> **Q2: run CEIL with access only to expert demonstrations.**
>
> **A2:** Thank you for the suggestion. In single-domain IL tasks (with only expert demonstrations), CEIL and BC (behavior cloning) are essentially the same, since Equation 5 is all about fitting expert demonstrations. However, if we consider cross-domain IL settings (with expert source and target data), CEIL can effectively use the source domain data to learn the contextual behavioral policy, and then use the expert target data to fit $z*$. Intuitively, although the source data is expert in the source domain, it can also be viewed as sub-optimal data for the target environment, thus CEIL can effectively utilize the cross-domain data to improve the performance. However, simple BC cannot do this.
>
> Here, we also run CEIL with access only to expert-level behaviors in the offline cross-domain IL settings and compare it to three baselines: 1) *BC over target* denotes that we train BC agent only over the target data; 2) *BC over source+target* denotes that we train BC agent over the combined source and target data; 3) *BC Fine-tuning* denotes that we first train BC agent over the source data and then fine-tune the agent over the target data.
>
> |	|BC over target	|BC over source+target	|BC Fine-tuning	|CEIL|
> |  ----  | ----  | ----  | ----  | ----  |
> |Hopper	|46.5 	|37.8 	|69.9 	|**93.7** |
> |Halfcheetah	|15.4 	|11.5 	|44.3 	|**48.1** |
> |Walker2d	|97.6 	|92.5 	|104.2 	|**111.8** |
> |Ant	|72.6 	|72.0 	|83.1 	|**95.0** |
>
> We can see that both *BC over source+target* and *BC over target* perform poorly (with 5 expert demonstrations in target), and *BC Fine-tuning* can slightly improve the performance. Our CEIL can effectively achieve the best performance.
>
>
> Thanks again for all of your constructive suggestions, which have helped us improve the quality and clarity of our paper.

---

> > ### Comment · Reviewer_vVku · 2023-08-12
> > **Thanks for the additional experiments!**
> >
> > I have read the authors' rebuttal and the other reviews, and am maintaining my score of 7.

---

### Author Rebuttal · Authors · 2023-08-08

Dear reviewers,

Thank you for all of your constructive suggestions, which have helped us improve the quality of our paper.

This general response provides a summary of the experimental requirements you suggested. Below, we list the content of each chart presented in the submitted PDF.
+ Figure 1: New experimental results in Atari and Adroit domains. [Reviewers vVku and 2c9A.]
+ Figure 2 and Table 1: Ablation studies on the optimization of $f_\phi$ and the objective of J_MI. [Reviewers PiwE and 2c9A respectively.]
+ Figure 3: Adding exploration bonus for C-on-LfO tasks. [Reviewers 7vt9 and 2c9A.]
+ Table 2: Comparison with respect to transformer-style methods. [Reviewer STjS.]

Thanks again for reviewing our submission, we do not take it for granted. We hope we have resolved all of your concerns. We are always willing to answer any of your further concerns.

---

### Decision · Program_Chairs · 2023-09-21

**Decision:**

Accept (poster)

**Comment:**

Dear Authors,

After careful evaluation and rigorous discussions with the reviewers, we recognize that the submitted paper presents a method poised to make a significant contribution to the IL field. The methodology, by updating the embedding function of a contextual variable, an optimal contextual variable, and a policy conditioned on that variable, showcases a novel approach. Particularly praiseworthy are the experimental results, highlighting improved performance across various IL tasks. The paper's breadth, covering eight different IL settings, and its depth, outperforming previous baselines in most environments, reflects its robustness. The empirical analysis provides valuable insights, further strengthening the research's practical significance. Furthermore, the paper's quality in terms of clarity, extensive review, and detailed implementation details in the appendix distinguishes it from other submissions.

Nevertheless, there are aspects of the work that merit attention. One of the most pressing concerns is the introduction of several hyperparameters without clear guidance on their selection. In addition, ambiguities persist in the representation of regularization losses, making it difficult for readers to follow without unnecessary confusion. The disparity between the theoretical and practical objectives, especially concerning the optimization of the hindsight embedding function, raises questions about the methodology's consistency. While the paper claims to address all online, offline, LFD, LFO, and cross-domain issues, the provided experimental evidence seems limited to back this broad claim comprehensively. Insights into the effectiveness of the J_MI term and its alignment with the expert demonstration in the latent space would be instrumental. Similarly, the inclusion of comparisons with recent transformer-style methods and an extensive discussion on CEIL's superiority over existing GAIL-style works would bolster the paper's overall impact.

In light of the above, while the paper demonstrates substantial promise and significant advancements in IL, addressing the aforementioned issues will enhance its value in the community. We hope that the feedback aids in the paper's refinement, making it an essential read for all researchers in the IL domain.

Best regads,

Area Chair